

# Validity and limitations of simple reaction kinetics to calculate concentrations of organic compounds from ion counts in PTR-MS

Rupert Holzinger[1], W. Joe F. Acton[2], William J. Bloss[3], Martin Breitenlechner[4], Leigh R. Crilley[3], Sébastien Dusanter[5], Marc Gonin[6], Valerie Gros[7], Frank N. Keutsch[4], Astrid Kiendler-Scharr[8], Louisa J. Kramer[3], Jordan E. Krechmer[9], Baptiste Languille[7], Nadine Locoge[5],Felipe Lopez-Hilfiker[6], Dušan Materić[1], Sergi Moreno[10], Eiko Nemitz[11], Lauriane L.J. Quéléver[12], Roland Sarda Esteve[7], Stéphane Sauvage[5], Simon Schallhart[13,a], Roberto Sommariva[3], Ralf Tillmann[8], Sergej Wedel[8], David R. Worton[10], Kangming Xu[1], Alexander Zaytsev[4]

[1] Institute for Marine and Atmospheric Research, Utrecht, IMAU, Utrecht University, the Netherlands.
[2] Lancaster Environment Centre, Lancaster University, Lancaster, LA1 4YQ, UK.
[3] School of Geography, Earth and Environmental Sciences, University of Birmingham, Birmingham, B15 2TT, UK.
[4] John A. Paulson School of Engineering and Applied Sciences, Harvard University, Cambridge, MA 02138, USA
[5] IMT Lille Douai, Université Lille, SAGE - Département Sciences de l'Atmosphère & Génie de l'Environnement, F-59000
[6] TOFWERK AG, Switzerland
[7] LSCE, Laboratoire des Sciences du Climat et de l'Environnement, Unité Mixte CEA-CNRS-UVSQ, IPSL, CEA/Orme des Merisiers, 91191 Gif-sur-Yvette, France.
[8] Institute of Energy and Climate Research, IEK-8: Troposphere, Forschungszentrum Jülich GmbH, Jülich, Germany
[9] Aerodyne Research Inc. Billerica, MA, 01821 USA
[10] National Physical Laboratory, Hampton Road, Teddington, Middlesex, TW11 0LW, UK.
Lille, France.
[11] Centre for Ecology & Hydrology, CEH, Bush Estate, Penicuik, Midlothian, EH26 0QB, UK.
[12] Institute for Atmospheric and Earth System Research - INAR/ Physics, P.O. Box 64, FI-00014 University of Helsinki, Finland.
[13] Finnish Meteorological Institute, P.O. Box 503, FI-00101 Helsinki, Finland.

[a] now at: Finnish Meteorological Institute, Helsinki, Finland

*Correspondence to*: Rupert Holzinger (r.holzinger@uu.nl)

**Abstract.** In September 2017, we conducted the Proton-transfer-reaction mass-spectrometry (PTR-MS) Intercomparison campaign at CABauw (PICAB), a rural site in central Netherlands. Nine research groups deployed a total of eleven instruments covering a wide range of instrument types and performance. We applied a new calibration method based on fast injection of a gas standard through a sample loop. This approach allows calibrations on time scales of seconds and within a few minutes an automated sequence can be run allowing to retrieve diagnostic parameters that indicate the performance status. We developed a method to retrieve the mass dependent transmission from the fast calibrations, which is an essential characteristic of PTR-MS instruments, limiting the potential to calculate concentrations based on counting statistics and simple reaction kinetics in the reactor/drift tube. Our measurements show that PTR-MS instruments follow the simple



reaction kinetics if operated in the standard range for pressures and temperature of the reaction chamber (i.e. 1-4 mbar, 30-120 ℃, respectively), and a reduced field strength $E/N$ in the range of 100-160 Td. If artefacts can be ruled out, it becomes possible to quantify the signals of uncalibrated organics with accuracies better than ± 30 %. The simple reaction kinetics approach produces less accurate results at $E/N$ levels below 100 Td, because significant fractions of primary ions form water

hydronium clusters. De-protonation through reactive collisions of protonated organics with water molecules need to be considered when the collision energy is a substantial fraction of the exoergicity of the proton transfer reaction, and/or if protonated organics undergo many collisions with water molecules.

## 1 Introduction

During the last 20 years, PTR-MS developed to be a relevant and widely applied technique that resulted in major advances in the field of atmospheric sciences (e.g. Lelieveld et al., 2001; Kirkby et al., 2011; Park et al., 2013; Tröstl et al., 2016). Since the conception of PTR-MS, there has been awareness for the potential of this technique to provide quantitative measurements for compounds that cannot be calibrated (e.g. Hansel et al., 1999). However, in practice, this potential cannot be fully exploited without reliable and applicable methods to retrieve the mass dependent transmission of PTR-MS

instruments. There are valuable and highly cited publications that explore best practices in PTR-MS measurements (e.g. Blake et al., 2009; De Gouw and Warneke, 2007), including methods to calibrate and retrieve the transmission (Taipale et al., 2008). However, these methods are slow and labour intensive, and typically not included in an automated workflow. Therefore, calibrations and transmission retrievals are performed not frequently enough and as a consequence the long-term accuracy of PTR-MS measurements is often limited. As a result, PTR-MS is mainly used in campaign scale deployments and

there are only few long-term studies that cover more than 1-2 months of measurements (e.g. Holzinger et al., 2006), and to the best of our knowledge there is only one group performing multi-year PTR-MS measurements (Taipale et al., 2008).

In the context of the European funded ACTRIS program[1], we aim at establishing PTR-MS as a technique for long-term monitoring of trace gases. This requires a standard operation protocol (SOP) that includes effective calibrations and assures highest possible and controllable data quality. Despite more than two decades of PTR-MS measurements there are no

comprehensive inter-comparison studies and no SOPs that are widely used within the PTR-MS-user community. The quality of PTR-MS data largely depends on the skills and knowledge of the operator. In order to set first steps towards a widely accepted SOP, we organised the PTR-MS Intercomparison Campaign at CABauw (PICAB), which was performed in September 2017 at the CESAR observatory[2], a rural site near the village of Cabauw in central Netherlands. The campaign was conducted under the auspices of the European infrastructure program ACTRIS-2 and attracted 9 groups from Europe

---

[1] https://www.actris.eu/
[2] http://www.cesar-observatory.nl/



and the United States with a total of 11 PTR-MS instruments, including latest developments of the technology such as PTR3 (Breitenlechner et al., 2017) and Vocus (Krechmer et al., 2018) instruments.

In this work, we investigate the power and limitations of a simple reaction kinetics based calibration approach, drawing on the results obtained with a novel calibration method based on injections of a gas standard from a sample loop. These

calibrations have been applied several times on all instruments participating at the campaign. Table 1 gives an overview of the 10 instruments which were used for this study. Details about the experimental setup are given in the method section. In addition, we frame methods on how to retrieve compound specific measured sensitivities and the instrument specific transmissions from the gas standard injections, and how to calculate the expected compound specific sensitivities using a simple reaction kinetics model. In the results section we discuss the wide range of measured sensitivities before exploring in

depth the agreement between measured and expected sensitivities. We were able to constrain limitations of the method. Several artefacts and clear directions for future work became apparent from the analysis.

## 2 Method

### 2.1 The calibration unit

The calibration unit is depicted in Figure 1. The core piece is a 250 µL sample loop connected to a 6-port valve allowing well defined gas standard injections into the PTR-MS instruments. A small flow (~10 mL/min) of carrier gas transports the content of the sample loop to a T-connection where it is mixed into a larger flow (0.2-2 L/min) of dry or humidified carrier gas. The small flow is tuned to produce a pulse duration of approximately 1 second. The larger flow is used for diluting the gas standard to ~5 nmol/mol for the Vocus, ~2 nmol/mol for the PTR3 (additional extra dilution), and to ~50 nmol/mol for

all other instruments. After dilution the mixing ratios are large enough to ensure good counting statistics, but also small enough to avoid saturation effects. In the supplemental Figure S1, we show the standard addition protocol. In essence, the sequence consisted of 50 sample loop injections, i.e. 5 sets of 10 injections using dry nitrogen, dry air, humidified air, humidified nitrogen, and dry nitrogen as carrier gas, respectively. On selected instruments and occasions, calibrations were performed sequentially at different $E/N$ values to investigate their effect on the calibrations. During the campaign, we used

two different gas standards produced by Apel-Riemer Environmental, Inc., USA (AR), which was used until September 22, 2017, and the National Physical Laboratory, UK (NPL), which was used from September 23 onwards. Both gas standards contained compounds that are entirely detected at their protonated mass as well as a few compounds that partially fragment during protonation (e.g. monoterpenes, siloxanes, and isoprene). The compounds in the gas standards cover the mass to charge ($m/Q$) range 33-373 Th. Details on the gas standards are given in the supplemental Tables S1 and S2. Figure 2 shows

an example of the raw signal (in counts per second, cps) during the 50 injections of the calibration on the 'TOFqi LIL' instrument on September 21, 2017, which is representative for all calibrations and instruments. The 16 compounds in the gas





standard produce 22 ions which are all shown in the panels of Figure 2. The different colours indicate the carrier gas. The reproducibility was tested by totalling the signal of individual injections and calculating the standard deviation of the 10 injections using the same carrier gas. This analysis showed that the reproducibility was typically 1% unless counting statistics were the limiting factor (see percent values printed in the charts of Figure 2).

## 2.2 Data processing

All basic data processing of PTR-MS instruments with a Time Of Flight (TOF) mass analyser was done with PTRwid (Holzinger, 2015). For the subsequent analyses we used the raw data output files that provide time-series of ion signals (in cps) for all ions that were auto-detected in the mass spectrum. These raw data closely correspond to the raw data obtained
from PTR-MS instruments using a quadrupole mass spectrometer (QMS), allowing TOF and QMS instruments to be directly compared.

### 2.2.1 Retrieving the transmission

We modelled the transmission as a combination of three functions with a total of five parameters. We developed an algorithm to retrieve the five parameters from gas standard injections as shown in Figure 2. The three functions optimised the transmission in the medium, low and high $m/Q$ range, respectively:

1)  The characteristic of the mass spectrometer in the medium $m/Q$ range (59-122 Th) according to Eq 1:

$$f_1(M) = M^a ; \qquad (1)$$

where $M$ corresponds to $m/Q$, and $a$ is a parameter between 0 and 2 describing the characteristics of the mass analysers in the medium $m/Q$ range. To optimise this parameter, we used the signal of compounds in the gas standard that are detected in the range 59-122 Th. For TOF instruments $a$ is expected to be around 0.5, because the kinetic energy of the ions is proportional to the square of their velocity, whereas QMS instruments should exhibit little to no mass dependent discrimination in this $m/Q$ range and thus $a$ is expected to be close to 0.

The retrieval algorithm calculates an initial value for the parameter $a$, by calculating a linear fit of the function $f(M) = S_m \times M^a$, where $S_m$ is the measured sensitivity (see below). The condition for the initial value for $a$ ( in the range 0 to 2) is that the linear fit function of $f(M)$ produces a zero slope. For fragmenting compounds (e. g. isoprene) we added the measured sensitivity of the fragment and the protonated ion.

2)  A 'high masses pass' filter according to Eq 2:




$$f_2(M) = \frac{1}{1 + e^{-\frac{M - M_L}{w_L}}} \; ; \qquad\qquad (2)$$

where $M$ corresponds to $m/Q$, $M_L$ is the $m/Q$ around which the 'high masses pass' filter becomes active, and $w_L$ is the filter slope at $M_L$. This filter is used to model the reduced transmission in the low $m/Q$ range that mostly results from the ion optics between drift tube/reactor and mass analyser. The parameters $M_L$ and $w_L$ are optimised by optimising the agreement between $S_m$ and $S_{expd}$ (see below) for all compounds in the gas standard that are detected below 60 Th.

The retrieval algorithm optimised the parameters $a$, $M_L$ and $w_L$ in an 2-step loop. In step 1 the parameters $M_L$ and $w_L$ were optimised as described above, with the condition that $f_2(M>60) > 0.98$. In step 2, the parameters $a$ was optimised to maximize the agreement between measured and expected sensitivities for compounds detected in the range 59-138 Th. For fragmenting compounds (isoprene, monoterpenes) we added the measured sensitivity of the fragment and the protonated ion.

   3)   A 'low masses pass' filter according to Eq 3:

$$f_3(M) = \frac{1}{1 + e^{\frac{M - M_H}{w_H}}} \; ; \qquad\qquad (3)$$

where M corresponds to $m/Q$, $M_H$ is a parameter that sets the $m/Q$ around which the 'low masses pass' filter becomes active, and $w_H$ is the filter slope at $M_H$. This filter is used to model the reduced transmission in the high $m/Q$ range, which can be changed by ageing of the microchannel plate or secondary electron multiplier in TOF and QMS analysers, respectively. In order to optimise the parameters $M_H$ and $w_H$ we use all compounds in the gas standard that are detected above 120 Th.

The retrieval algorithm optimised the parameters $M_H$ and $w_H$ by maximizing the agreement between measured and expected sensitivities for compounds detected above 120 Th, with the condition that $f_3(M<137) > 0.98$. For fragmenting compounds (monoterpenes, D3, D4, and D5) we added the measured sensitivity of the fragment and the protonated ion.

Finally, the transmission is calculated by multiplying Eq 1, 2, & 3 and a normalization step to set the transmission at 59 Th to 1 (Eq 4). We chose to normalize to 59 Th (protonated acetone) because the transmission around this $m/Q$ is high for all mass analysers used in this study, besides from that it is an arbitrary choice:

$$\tau(M) = \frac{f_1(M) f_2(M) f_3(M)}{f_1(59Th) f_2(59Th) f_3(59Th)} \qquad\qquad (4)$$

Note that the algorithm considers $H_3O^+$ and $H_2OH_3O^+$ as primary ions that both protonate with the same efficiency. However, the protonation efficiency of hydronium water clusters is reduced for many compounds, and therefore we expect the best results for measurements with low contributions of water hydronium clusters to the total primary ion signal.

For the PTR3, the number of species which can be used for this approach is limited to those where de-protonation reactions are negligible and the protonation efficiencies for hydronium and hydronium water clusters are similar (more details in



appendix A). Therefore, only six species were taken into account for retrieving the transmission for this instrument as shown in Figure A2: Acetone, MVK, MEK, and the three siloxanes. For this study we used the first 10 injections with dry $N_2$ as carrier gas to retrieve the transmissions, and the remaining 40 injections using dry and humidified air and $N_2$ for validation. A few example retrievals are shown in Figures S2 and S3.

### 2.2.2 Retrieving and calculating sensitivities

We report three types of sensitivity:

1. The measured sensitivity as it is actually measured in the field. The measured sensitivity $S_m$ is often reported in units of cps/ppb[3] and can be expected from every single injection according to Eq. 5:

$$S_m(x) = \frac{C(x) \times q_V}{n(x)} \; ; \qquad (5)$$

where $C(x)$ is the total signal (*counts*) of compound $x$ measured during an injection, $q_V$ is the total flow provided by the calibration unit in moles/s, $n$ is the amount of substance of compound $x$ in the sample loop in moles. $C(x)$ is calculated by totalling the signal of $x$ during an injection and subtracting a baseline signal that is recorded before and after the injection[4], and $n(x)$ is calculated according to Eq. 6:

$$n(x) = \frac{c(x) \times V_l \times p_l}{T_l \times R} \; ; \qquad (6)$$

where $c(x)$ is the fraction of compound $x$ in the gas standard in mol/mol, $V_1$ is the volume of the sample loop in $m^3$, $T_l$ and $p_l$ are the temperature and pressure in the sample loop in K and Pa, respectively. The parameter $R$ correspond to the gas constant (8.31 $J.mol^{-1} K^{-1}$).

The measured sensitivity $S_m(x)$ is a direct proxy of the statistical uncertainty. Together with the instrumental background this quantity determines the precision and the limit of detection.

2. The normalized sensitivity ($S_N$) is calculated similarly. The only difference is the multiplication by a dimensionless factor $N$ that normalizes to a reagent ion flux of $10^6$ *cps* and corrects for the transmission:

---

[3] The unit of equation 5 is (counts*moles)/(moles*s). To express sensitivities in cps/ppb, the trivial relation 1 mol/mol = $10^9$ nmol/mol is used.

[4] We totalled the signal by considering only the main isotopologue of the protonated ion and the fragments (41 Th for isoprene, 41 and 69 Th for MBO, 81 Th for monoterpenes, and 207, 281, and 355 Th for D3, D4, and D5, respectively). These signals were background corrected and multiplied by a factor to account for the signal that is expected on the m/Q of the minor isotopologues (i.e. molecules containing D, $^{13}C$, or $^{18}O$).





$$S_N(x) = \frac{C(x) \times N \times q_V}{n(x)} ; \tag{7}$$

If we consider $H_3O^+$ and $H_2OH_3O^+$ as primary ions, the factor N is calculated according to Eq. 8:

$$N = \frac{10^6\, cps}{F(H_3O^+)\dfrac{\tau(x)}{\tau(H_3O^+)} + F(H_2OH_3O^+)\dfrac{\tau(x)}{\tau(H_2OH_3O^+)}}, \tag{8}$$

where $F(H_3O^+)$ and $F(H_2OH_3O^+)$ are the fluxes[5] of the $H_3O^+$ and $H_2OH_3O^+$ primary ions in cps. The functions $\tau(\dots)$ are the

transmission efficiencies at the $m/Q$ of reagent and product ions, respectively, as defined in Eqs. 1 to 4.

The normalized sensitivity ($S_N$) is a useful quantity that can be related to fundamental kinetic parameters in the PTR-MS. Different instruments that operate under similar conditions (i.e. pressure, temperature, humidity and electrical field across the drift tube) should retrieve similar normalized sensitivities.

3.   Based on simple reaction kinetics the expected sensitivity can be calculated according to Eq. 10:

$$S_{expd}(x) = k(x) \cdot F \cdot t \cdot n_R = k(x) \cdot F \cdot t \cdot n_0 \frac{p_R T_0}{T_R p_0} \tag{10}$$

where $t$ is the residence time and $F$ the flux of the reagent ions ($H_3O^+ + H_2OH_3O^+$) in the reaction chamber (drift tube), $p_R$, $T_R$, and $n_R$ are pressure, temperature, and gas density in the reaction chamber, the constants $p_0 = 101325$ Pa, $T_0 = 273.15$ K, and $n_0 = 2.7 \times 10^{19}$ molecules cm$^{-3}$ are pressure, temperature and number density of air under normal conditions,

respectively. The reaction rate constant, $k(x)$, is in the range 1.85- $3.39 \times 10^{-9}$ cm$^3$ s$^{-1}$ molecule$^{-1}$ for all compounds present in the gas standards. The values that were used in this study are given in Table S3. Note that the expected sensitivity can be directly compared to the normalized sensitivity ($S_N$) if we use $F = 10^6$ cps for the reagent ion flux. For the 'PTR3 HAR' instrument the residence time, $t$, is given by the flow through the reaction chamber and has been estimated to be $3.5 \pm 0.5$ ms for all calibration measurements. For the other instruments, the residence time has been calculated according to Eq. 11:

$$t = \frac{1}{K}\frac{d}{E} = \frac{1}{K_0}\frac{p_R T_0}{p_0 T_R}\frac{d}{E}, \tag{11}$$

where $d$ is the length, and $E$ the electrical field strength across the length of the reaction chamber. The constant $K$ is the mobility of $H_3O^+$ ions and $K_0$ is the reduced mobility of $H_3O^+$ ions for which we used a value of 2.7 cm$^2$ V$^{-1}$ s$^{-1}$ (Dotan et al., 1976).

---

[5] Note that fluxes are a relative quantity here. We do not know the real ion flux in the drift tube, but we assume that the real ion flux is a fraction of the measured flux that only depends on $m/Q$ (i.e. the transmission ).





The normalized and expected sensitivities, $S_N$ and $S_{expd}$, can be directly compared and provide a measure on how well ionization in the PTR-MS is constrained by basic reaction kinetics, which is important to assess the accuracy of concentrations for compounds that are not calibrated frequently with a gas standard.

Note that Eq. 10 and 11 are analogous to methods presented by Hansel et al. (1995) that calculate the volume mixing ratio of VOCs based on kinetic conditions in the drift tube.

For the "PTR3 HAR", higher water hydronium clusters need to be considered and de-protonation is a non-negligible process for several species present in the calibration standard[6]. A conceptual framework for calculation of sensitivities taking this process into account is presented in Appendix A. Figure A1 shows that calculation for compounds with high proton affinity – where de-protonation is negligible – leads to accurate (within 20%) predictions. For species which show significant de-protonation rates, on the other hand, calculated sensitivities have increasing uncertainties and water dependencies.

## 3 Results and discussion

### 3.1 Retrieved transmissions

All transmissions shown in Figure 3 have been retrieved from the first 10 injections that used dry $N_2$ as carrier gas. Transmissions obviously vary between instruments but also over time for individual instruments. However, instruments that were operated under constant conditions (e.g. TOF8000 UHEL, and QMS LSCE) exhibited little variation in transmission over time. We find that typically 'flatter' transmissions were retrieved when the instruments were (deliberately) operated at lower $E/N$ (thin lines in Figure 3). Considering that higher water clusters of the hydronium ion ($H_3O^+(H_2O)_n$, with $n > 1$) could provide significant fractions of the primary ion signal at lower $E/N$ (which is not accounted for in our algorithm), we would expect to retrieve 'steeper' transmissions in the range 20-50 Th. However, we do not observe this effect, so we conclude that we did not miss a significant fraction of the primary ion signal. On the other hand, flatter transmissions were also retrieved from gas standard injections that used humidified carrier gas. An example is shown in Figure S3. This suggests that several compounds in the gas standard must be detected with lower sensitivity than expected. The cause for reduced sensitivities includes slower proton transfer with hydronium clusters, as reported for benzene (Warneke et al., 2001), as well as more complicated ion chemistry involving back reaction of protonated compounds with water vapour as has been reported for formaldehyde (Hansel et al., 1997). Together with evidence presented below we suggest that the flattening of transmission with lower $E/N$ is caused by slower proton transfer with hydronium clusters.

---

[6] Note that the E/N of PTR3 instruments is typically in the range 60-90 Td. Partly this is to reduce the influence of backward reactions (de-protonation), which are important for several compounds because of the many collisions between ions and molecules in the PTR3 reactor.



## 3.2 Measured and expected sensitivities

Figure 4 shows that the measured sensitivities for all instruments and compounds in the gas standards cover the range 1-
$2\times10^5$ cps/ppb. Note that identifying the "best" instrument was not the purpose of this study. Some instruments were
deliberately operated outside the optimal range in terms of tuning (sensitivity) or energetics in the drift tube/reactor (E/N). In

general, we note that the large difference in sensitivity for the PTR-MS instruments is rooted in different tuning and ion
optics, or innovative concepts that further boost the sensitivity of the Vocus and PTR3 instruments. For many compounds the
PTR3 instrument is at least one order of magnitude more sensitive than any other instrument. This is due to the very different
conditions under which the PTR3 instrument is operated: the PTR3 instrument gains sensitivity by allowing for longer
reaction times and a higher pressure in the reaction chamber rather than by boosting the primary ion signal. As a result,

reagent ions undergo approximately 1000 times more collisions with the analyte gas molecules compared with the other
instruments. While this concept overall leads to greatly enhanced sensitivities, it also complicates quantification: de-
protonation reactions of the form $RH^+ + H_2O \rightarrow H_3O^+ + R$ limit the sensitivity for a broader range of species, while in
other instruments this is only the case for formaldehyde and a few other compounds with proton affinities just slightly above
that of water. Furthermore, the PTR3 is operated at a reduced electric field strength of $60 \pm 5$ Td, therefore the primary ion

distribution is dominated by water hydronium clusters. Thus, ligand switching reactions with internal proton transfer
dominate over direct proton transfer from the hydronium ion. Both effects lead to relatively poor and uncertain sensitivities
for compounds having a low proton affinity and/or low dipole moment, both preventing efficient ligand switching reactions.
This explains that the measured sensitivities of the different compounds cover several orders of magnitude for the PTR3
instrument, whereas for all other instruments the measured sensitivities are typical within one order of magnitude (Figure 4).

The lower sensitivity of the 'QMS LSCE' instrument for higher $m/Q$ values is a property of the quadrupole mass analyser
that is used in this instrument. Figure 4 reveals lower than expected sensitivities in the Vocus instrument for methanol,
acetonitrile, acetaldehyde, 3F-benzene and 3Cl-benzene. For the three lighter compounds the reason is a very sharp 'high
mass pass' filter[7] that suppresses virtually the entire signal of protonated methanol at $m/Q = 33$ Th, and therefore we exclude
Vocus methanol data from further analysis. The filter reduces protonated acetonitrile (42 Th) and acetaldehyde (45 Th) by

about 90 %, however, this should be accounted for by the retrieved transmissions.
Further insights can be obtained from looking at the ratio of measured to expected sensitivity which should be unity if the
reaction kinetics are accounted for correctly and if there are no additional losses. Figure 5 shows this ratio for all compounds
in the gas standard and for all instruments. Data from all injections using dry carrier gas ($N_2$ or air), except those that were
used to retrieve the transmissions, are included in Figure 5. For many compounds the ratio deviates much less than $\pm$ 30 %

from unity; the boundary of this range is indicated by the black horizontal lines in Figure 5. In principle, this demonstrates

---

[7] Position and sharpness of the filter are adjustable. The default factory settings aim at optimizing the detector lifetime.



the potential of PTR-MS to quantify organic compounds without calibration. However, some limitations emerge from the data shown in Figure 5:

1. Above 150 Th spreading between instruments becomes larger. The likely reason for this is that transmissions are less constrained in this range. Most calibrations were done with the NPL gas standard that contained only two compounds above $m/Q$ = 150 Th (D4 and D5 siloxanes). There are indications that these two compounds are sticky (note that Figure 2 shows a poorer reproducibility of these compounds) and thus vulnerable to surface artefacts. Moreover, the combination of Equations 1, 2 and 3 may not be the best choice to replicate the real behaviour of all of the mass analysers used. The latter is clearly the case for quadrupole mass analysers, when considering the D4 siloxane ratio for the 'QMS LSCE' instrument (low blue point at 300 Th in Figure 5). Many instruments show a dipole between D4 and D5 siloxanes (D4 low and D5 high), because the ratios measured to expected sensitivity were inconsistent with the spectrum of retrievable transmissions (dictated by Equations 1-3). Such a case is shown in Figure S2a, where the algorithm minimized the error by distributing the inconsistency between D4 (too low) and D5 (too high), whereas Figure S2b shows a case with ratios consistent with possible transmissions.

   These issues are likely resolvable with an improved gas standard that contains more compounds in the range 150-400 Th.

2. Methanol is detected with a lower sensitivity than expected in most instruments. A close inspection revealed that injections using humidified carrier gas clearly produced higher signals and the injections using dry carrier gas exhibited significant tailing. Both features are visible in the top left chart of Figure 2. We suggest that this issue is caused by wall effects in the instruments and/or their inlet lines and that the issue is less pronounced under humidified conditions. A similar issue, but less pronounced, was observed for MVK and MEK for the 'TOF8000 UHEL' instrument. These features demonstrate that surface effects in the PTR-MS instruments and their inlet systems can jeopardize quantitative detection of organic compounds.

3. For the Vocus instrument the ratio measured to expected sensitivity was biased high for acetonitrile and low for acetaldehyde, isoprene, benzene, 3F-benzene, and 3Cl-benzene. The high bias of acetonitrile may be an artefact of the transmission algorithm that tried to compensate for the inconsistency caused by lower than expected sensitivity of acetaldehyde. Correcting the bias would further decrease the ratio obtained for acetaldehyde. With respect to other instruments, the Vocus is unique because the reaction chamber contains approximately 30% of water vapour. Therefore we suggest that lower than expected sensitivities of these compounds are the result of reactions of protonated compounds with water vapour. The higher $E/N$ compared to other instruments is another factor that helps to overcome the energy barrier for these reactions and makes de-protonation more efficient.



### 3.3 Can PTR-MS quantify uncalibrated organic compounds?

The results shown in Figure 5 suggest that, in principle, PTR-MS is able of quantifying compounds without calibration based on simple reaction kinetics and a correctly retrieved transmission, if surface effects and unknown fragmentation can be excluded. However, in addition to the aforementioned reservation, the dependence of the retrieved transmissions on $E/N$ is another concern. In this section, we will further discuss to what extent PTR-MS is capable to perform quantitative measurements of uncalibrated compounds. The results from all gas standard measurements and all instruments are shown in Figure 6 for acetone as an example. Similar figures for the other compounds are provided in the supplemental Figures S4-S18. First-row panels in Figure 6 show the ratio of primary ions $H_2OH_3O^+$ to $H_3O^+$ as well as operating conditions of the instruments, and the second-row panels show the measured sensitivities for acetone as displayed in Figure 4. The third-row panels in Figure 6 show that the normalized sensitivities, $S_N$, are within one order of magnitude (6-50 cps/ppb). This demonstrates that the reagent ion signal is the primary factor that determines the sensitivities of individual PTR-MS instruments. The bottom-row charts in Figure 6 show that measured and expected sensitivities typically agree within less than 10 % (standard deviation) for all instruments, with some exceptions visible for measurements that used humidified carrier gas.

The data shown in Figure 5 represent idealized conditions because the dry carrier gas supresses the production of water hydronium clusters. Such conditions cannot be achieved in many common applications of PTR-MS. Therefore, we show the ratio of measured to expected sensitivities for the humidified calibrations in Figure 7. The main message is that for many compounds the ratio is still within ± 30 % of unity, however, the spreading between instruments is larger compared with dry gas standard injections. For some instruments the spreading between individual measurements is increased as well (error bars in Figure 7 compared with error bars in Figure 5). A closer inspection of Figure 7 reveals the following:

1. We observe no significant changes for the Vocus instrument. This is expected because the humidified carrier gas does not add significant extra humidity to the 30 % water vapour that is present in the reactor anyway. Besides the Vocus, also the instruments 'TOF8000 UU', 'TOF8000 UHEL', 'QMS LSCE', and 'TOFqi BHAM' produce very similar results with dry and humidified carrier gas. These instruments were operated at relatively high $E/N$ values in the range 100-135 Td (except one measurement with 'TOFqi BHAM'). The chosen operating conditions for these instruments resulted in relatively low levels of water hydronium clusters, so that the expected sensitivities produced accurate results. The results for methanol even improved slightly showing that the surface effects are eased a bit under humid conditions. The same holds for MVK and MEK measured with 'TOF8000 UHEL'.

2. The ratios for benzene and 3F-benzene are lower. This is likely due to the well-documented fact that these compounds are not efficiently protonated by water hydronium clusters (Warneke et al., 2001).

3. The instruments 'TOFqi LIL', and to a lesser extent 'TOFqi CEH' were biased low by typically 10-30 % for all compounds except the siloxanes (D3, D4, and D5) and methanol for 'TOFqi LIL'.




4. For the instruments 'TOF1000 UU' and 'TOF8000 FZJ', the spread between individual gas standard measurements is much increased. A closer inspection revealed that measured and expected sensitivities were generally consistent for measurements done at *E/N* levels above 100 Td. However, measurements at *E/N* levels below 100 Td revealed significant inconsistencies between measured and expected sensitivities. This has also been the case for other instruments during occasional measurements at low *E/N* (see deviations from unity in bottom row charts of Figure 6, and Figures S4-S18). We note that the inconsistency at low *E/N* results in sensitivities that are measured lower than expected, except for the 'TOF800 FJZ' instrument where the opposite was observed.

Points 3 and 4 warrant further discussion. Figure 8 summarises the results of a comparison of dry and humidified gas standard injections. The panels in Figure 8 show the ratios of different parameters measured with humidified versus dry carrier gas. Data printed in red, yellow, and blue are the ratios of (i) the primary ion signal ($H_3O^+$ + $H_2OH_3O^+$) corrected by the transmission , (ii) the uncorrected $H_3O^+$ signal, and (iii) the measured sensitivities, respectively. The latter has been calculated as the mean measured sensitivity of a core set of compounds (acetonitrile, acetaldehyde, acetone, isoprene, MVK, MEK, xylene, TMB, and monoterpene) that all exhibited very similar trends (see Figures S4-S18). For all instruments that performed gas standard measurements at *E/N* levels below 100 Td we observed that the measured sensitivity decreased for the core set compounds when the carrier gas was humidified. The likely cause is a reduced reaction speed with water hydronium clusters, which could be taken into account in more advanced models to calculate the expected sensitivity. At humidified conditions and an *E/N* around 80 Td, less than a few percent of the primary ions are present as $H_3O^+$ (de Gouw et al., 2003), which is the likely reason for very low measured sensitivities of benzene (Figure S9) and 3F-benzene (Figure S13). Note that only the 'TOF1000 UU' instrument measured low fractions of $H_3O^+$ as expected (ratios F37/F19 are in the range 6-9 for humidified injections at 80 Td); in all other instruments the cluster distribution was not preserved during the transfer from the drift tube into the mass spectrometer (F37/F19 always lower than 1.5, see top panels in Figure 6).

The ratio of measured to expected sensitivity is not sensitive to the humidity of the sample if both, the measured sensitivity, and the transmission corrected primary ion signal, vary in the same way, i.e. the blue and red data overlap in Figure 8. For the reasons discussed above, this is not the case for measurements at low *E/N*. Another process that causes separation of red and blue data is best visible in the 'TOFqi LIL' chart of Figure 8. This chart clearly shows that the cause is not a change in the sensitivity, but that for unknown reasons the primary ion signal is recorded higher during humidified measurements. Since the uncorrected $H_3O^+$ signal (yellow data in Figure 8) is recorded higher in the 'TOFqi LIL' instrument as well, we reject the possibility that this may be caused by an artefact in the transmission retrieval. Instead, we think that for unknown reasons primary ions are extracted to the mass analyser more efficiently under humidified conditions. The 'TOFqi CEH' instrument shows a similar, but less pronounced effect. An opposite effect was observed for the 'TOF8000 FZJ' instrument: during humidified low *E/N* measurements the primary ion signal was recorded lower for unknown reasons, but the measured sensitivity did not decrease correspondingly.

In response to the question posed in this section we state the following:





(i)  Quantitative detection (better than ± 30 %) is possible for *E/N* values above 100 Td if artefacts associated with the transmission of primary ions can be ruled out. The reasons for the artefacts are not explored in this study, but they may be associated with ion optics in the interface between drift tube and mass spectrometer, or with surface ageing (coating) in this region. These artefacts can be detected by comparing gas standard additions using dry and humidified carrier gas, respectively. A required condition is that most of the primary ion signal is present as $H_3O^+$ ion, which may require controlling water leakage from the ion source. Alternatively, higher *E/N* values can be applied to suppress the formation of water hydronium clusters.

(ii)  Backward reactions can significantly reduce the sensitivity for compounds with a proton affinity relatively close to that of water. This effect is well known and studied for formaldehyde (Hansel et al., 1997), but can also affect the detection of other compounds in instruments/setups that allow for many collisions of protonated compounds with water molecules as it is the case for the PTR3 instrument (low E/N and high drift tube pressure, see Appendix A) and to a lesser extent for the Vocus instrument (due to high levels of water vapour in the reaction chamber).

(iii)  Reliable quantification for *E/N* values below 100 Td becomes more complicated because increasing fractions of the primary ions are present in the form of water hydronium clusters. For a number of compounds, this resulted in reduced sensitivities up to 50 % (see Figures S4-S18), and even larger reductions were observed for benzene and 3F-benzene.

We note that an improved kinetic ion chemistry model that accounts for the cluster distribution, different reaction rates with clusters, the humidity, and the back reaction can expand the limits of quantitative operation of PTR-MS. In this study we did not explore dissociative proton transfer reactions because in traditional PTR-MS applications that focus on volatile organic compounds fragmentation of compounds is the exception rather than the rule. However, there are indications that this changes dramatically in new fields of application such as the analysis of semi-volatile organic compounds, condensed organics, and dissolved organics (Holzinger et al., 2010; Eichler et al., 2015; Materic et al., 2017). A recent intercomparison study (Gkatzelis et al., 2018) revealed that operating PTR-MS at lower *E/N* values strongly reduces the fragmentation of these compounds, which likely will make measuring at lower *E/N* an appropriate choice, especially if the disadvantages of that can be compensated with an improved reaction kinetic model. Finally, we note that the mentioned new fields of applications mostly concern compounds in the range 150-400 Th, which highlights the need to better constrain the transmission in this *m/Q* range.

## 4 Conclusions

We provided an analysis of more than 70 measurements following our calibration protocol on 10 different PTR-MS instruments over a 10-day period in September 2017. We outlined a simple reaction kinetics model and found that this model



accurately predicts the sensitivities if no artefacts interfere and the instruments were operated at *E/N* levels above 100 Td. We observed three different artefacts: (i) surface retention of methanol (stickiness) in all instruments and to a lesser extent for MVK and MEK in one instrument, (ii) reduced detection of primary ions under humidified conditions at low *E/N* in one instrument, and (iii) enhanced detection of primary ions under humidified conditions in two instruments featuring a

quadrupole transfer system between drift tube and TOF analyser. These artefacts caused errors of order -50 %, +50 %, and -20 %, respectively. At lower *E/N* the accuracy of the simple reaction kinetics model is limited because higher fractions of water hydronium clusters are present. De-protonation reactions can be of concern if the collision energy approaches the exoergicity of the proton transfer reaction and/or protonated compounds undergo many collisions with water molecules. These conditions are of concern for the detection of formaldehyde in all instruments, benzene and 3F-benzene in the Vocus

instrument and several additional compounds in the PTR3 instrument. The used gas standards do not contain sufficient compounds to constrain the transmission in the *m/Q* range 150-400 well enough. New fields of applications such as the detection of semi-volatile organic compounds, condensed and dissolved organics mostly concern organics with molecular weights above 150 Da; therefore, it is desirable to develop gas standards with a good coverage of this *m/Q* range. Moreover, reduced fragmentation will warrant the operation at lower *E/N* levels for these new applications. Therefore, more advanced

reaction kinetics models will be useful developments. However, overall we can conclude that PTR-MS is capable to measure uncalibrated compounds with an accuracy of ±30 % conditionally no unknown fragmentation occurs and de-protonation reactions are of minor significance, i.e. the proton affinity of the analyte is high.

Acknowledgements

This project has received funding from the European Union's Horizon 2020 research and innovation programme (ACTRIS-2) under grant agreement No 654109 and by the Dutch NWO Earth and Life Science (ALW), project 824.14.002. W.J.F. Acton, M. Breitenlechner, L.R. Crilley, L.J. Kramer, J.E. Krechmer, F. Lopez-Hilfiker, E. Nemitz, L.L.J. Quéléver, S. Schallhart, R. Tillmann, S. Wedel, and A. Zaytsev acknowledge Trans-National-Access (TNA) travel funding from ACTRIS-2 (grant agreement No 654109). E. Nemitz further acknowledges the UK Natural Environment Research Council

(NERC) through grants NE/P016502/1 for instrument funding and NE/R016429/1 as part of the UK-SCaPE programme delivering National Capability. L.L.J. Quéléver and S. Schallhart acknowledge the Finnish Centre of Excellence program (Project no 307331). L.L.J. Quéléver thank the European Research Council (ERC-Grant no 638703-COALA). W.J.F. Acton has received funding from Natural Environment Research Council (UK) grant NE/N006976/1, Sources and Emissions of Air Pollutants in Beijing (AIRPOLL-Beijing).




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



Table 1: Overview of the PTR-MS instruments participating in the inter-calibration exercise.

| ID | Institution | Manufacturer, year of production | Mass Analyser | Operating conditions (pdrift, $E/N$, Tdrift) | Ion optics |
|---|---|---|---|---|---|
| TOF1000 UU | Utrecht University | Ionicon , 2016 | short TOF, Ionicon | 1.8-3.8 hPa, 80-135 Td 60 °C | Static lens ion optics |
| TOF8000 FZJ | Forschungszentrum Juelich, | Ionicon , 2007 | HTOF, Tofwerk | 2.4 hPa 80-120 Td 60 °C | Static lens ion optics |
| TOF8000 UHEL | University of Helsinki | Ionicon, 2008 | HTOF, Tofwerk | 2.3-2.5 hPa 130 Td 60-35 °C | Static lens ion optics |
| TOF8000 UU | Utrecht University | Ionicon, 2008 | HTOF, Tofwerk | 2.7-3.2 hPa 100-120 Td 80-120 °C | Static lens ion optics |
| TOFqi BHAM | University of Birmingham | Ionicon, 2017 | HTOF, Tofwerk | 3.8 hPa 80-130 Td 80 °C | Quadrupole ion guide |
| TOFqi CEH | CEH/Lancaster University | Ionicon, 2017 | HTOF, Tofwerk | 3.8 hPa 80-120 Td 80 °C | Quadrupole ion guide |
| TOFqi LIL | IMT Lille Douai | Ionicon, 2016 | HTOF, Tofwerk | 3.8 hPa, 80-140 Td, 70°C | Quadrupole ion guide |
| Vocus | TOFWERK/ Aerodyne Research | Tofwerk, 2017 | LTOF, Tofwerk | 1 hPa 140-170 Td 30° C | Quadrupole ion guide |
| PTR3 HAR | Harvard University | Harvard University, 2017 | LTOF, Tofwerk | 65 hPa 60 Td, 30 °C | Quadrupole ion guide |
| QMS LSCE | LSCE Laboratoire des sciences du climat et de l'environnement | Ionicon, 2010 | QMG 422, Balzers | 2.2 hPa 132 Td 60 °C | Static lens ion optics |





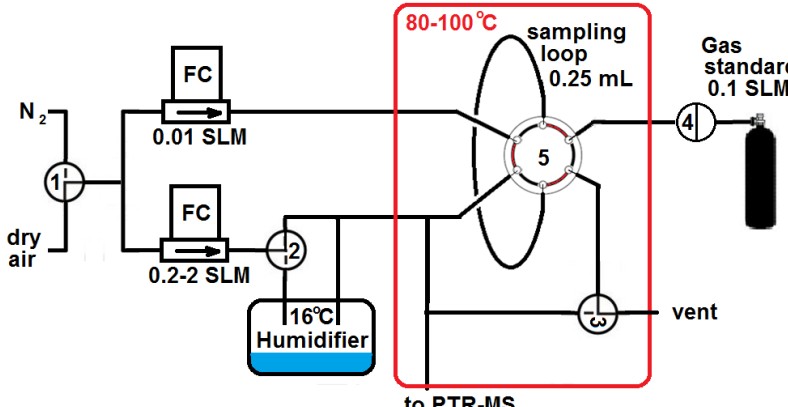

**Figure 1:** Schematic of the semi-automated calibration unit. Manual 3-way valves allow selection of dry air or nitrogen as carrier gas (valve 1), dry or humidified carrier gas (valve 2), and sample loop injections or dynamic mixing of carrier gas and gas standard (valve 3). Valve 4 (PFA-solenoid) and valve 5 (Valco 6-port with Restek sulfinert coating) are controlled to provide a sequence of 10 injections in one minute.



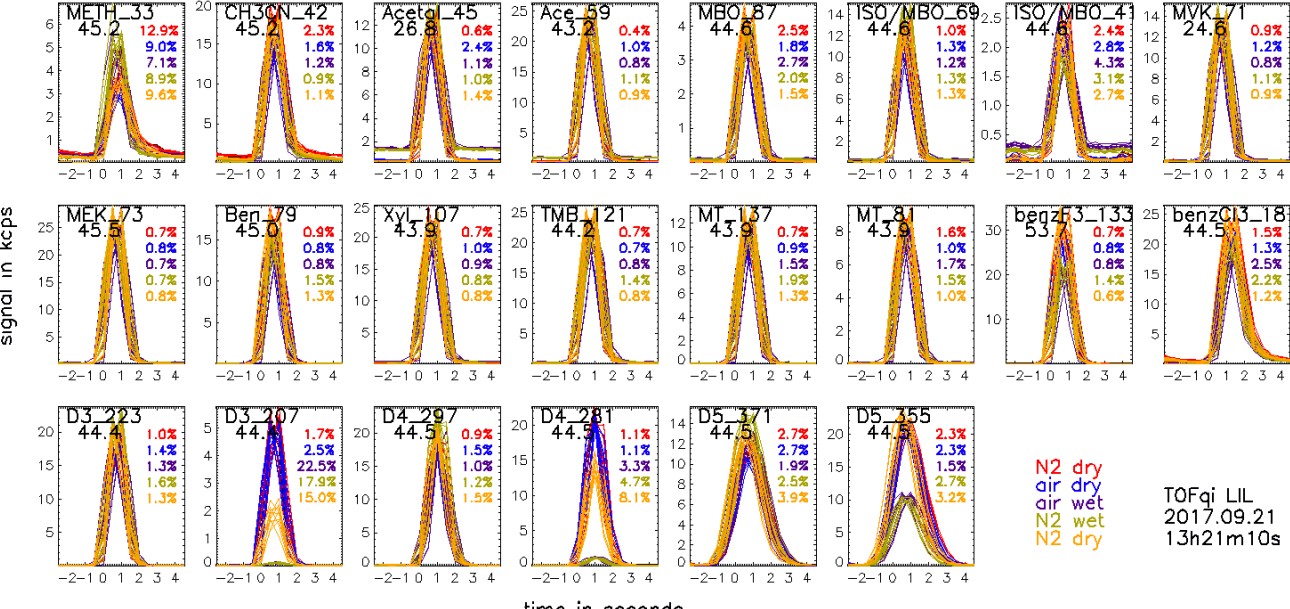

*Figure 2.* Raw count signal of the main ions produced by the organic compounds in the gas standard during the 50 sample loop injections. For example, the top right chart shows the signal at 71.049 Th originating from protonated methylvinylketone (MVK, C$_4$H$_6$O). In the top left of the chart an identifier code (including the integer *m/Q* value of the detected ion) is printed in black and the number below the identifier indicates the maximum volume mixing ratio (in nmol/mol) that is expected during an injection. The different colours correspond to the injections in different carrier gases (dry nitrogen, dry air, humidified air, humidified nitrogen, and dry nitrogen corresponding to red blue purple green, and yellow, respectively). The percent values printed at the right edge of each chart indicate the reproducibility of the 10 injections of each set.



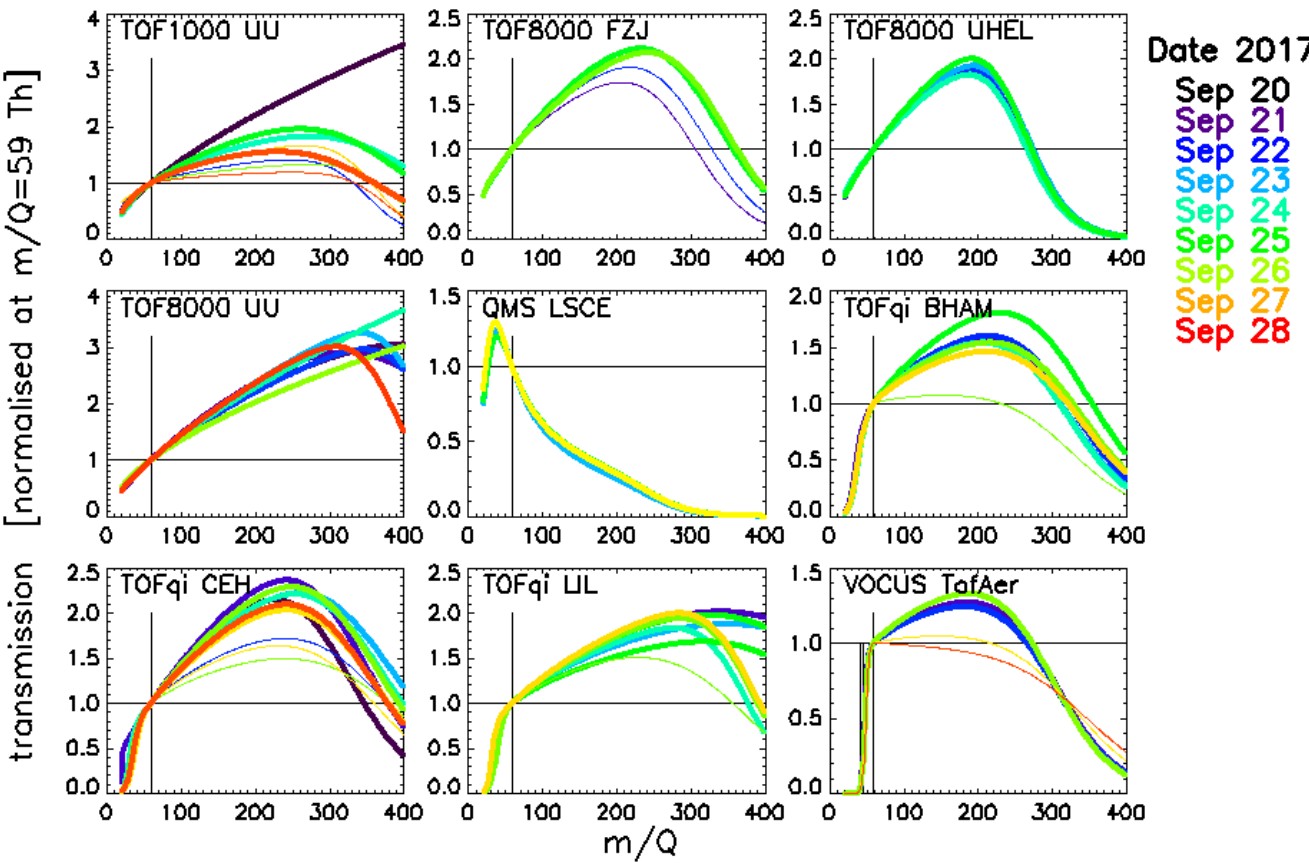

**Figure 3.** Retrieved transmission for all PTR-MS instruments (except the PTR3 instrument). Thin lines represent measurements at *E/N* below 150 Td, and 100 Td for the Vocus, and all other instruments, respectively. The colours indicate the date of the measurements.



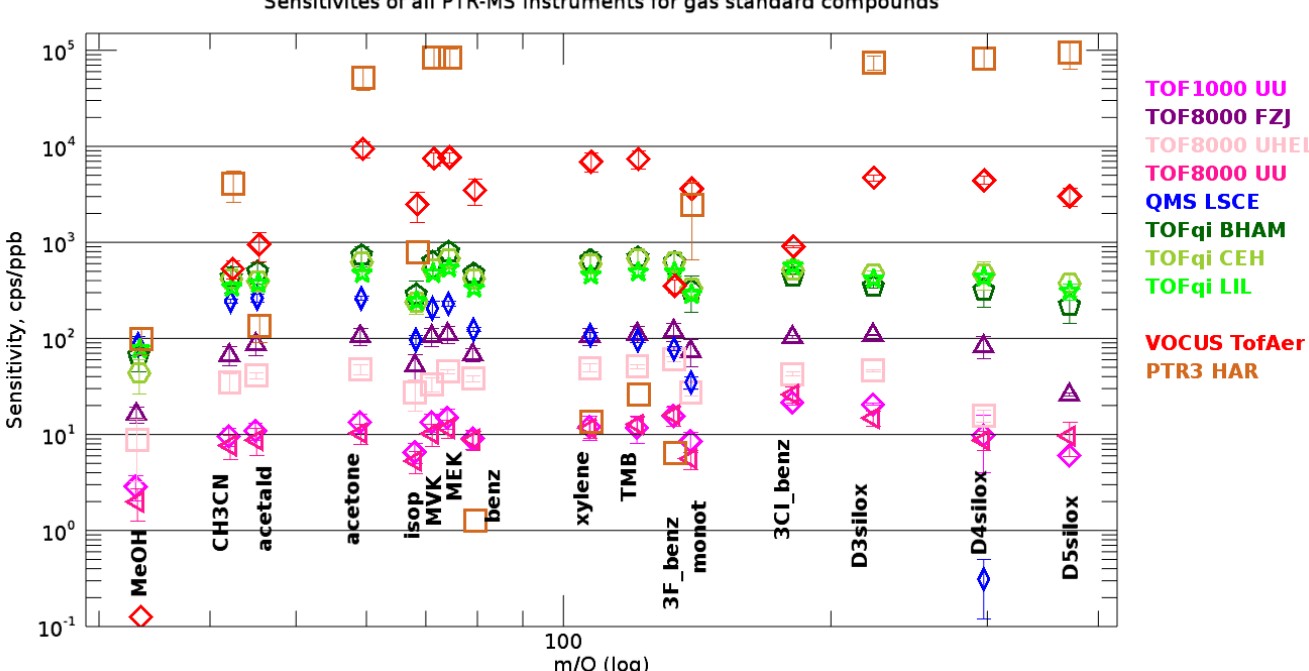

**Figure 4.** Mean measured sensitivities for all compounds in the gas standards and all PTR-MS instruments. The error bars represent the standard deviation of all calibrations with dry $N_2$ or air. The measured sensitivities cover more than 4 orders of magnitude. The compounds (protonated mass in parenthesis) from left to right are: methanol (33 Th), acetonitrile (42 Th), acetaldehyde (45 Th), acetone (59 Th), isoprene (69 Th)/methylbutenol (87 Th, main fragment on 69 Th), methylvinylketone (71 Th), methylethylketone (73 Th), benzene (79 Th), xylene (107 Th), trimethylbenzene (121 Th), trifluorobenzene (133 Th), 3-carene/α-pinene (137 Th), trichlorobenzene (181 Th), D3-siloxane (223 Th), D4-siloxane (297 Th), D5-siloxane (371 Th).





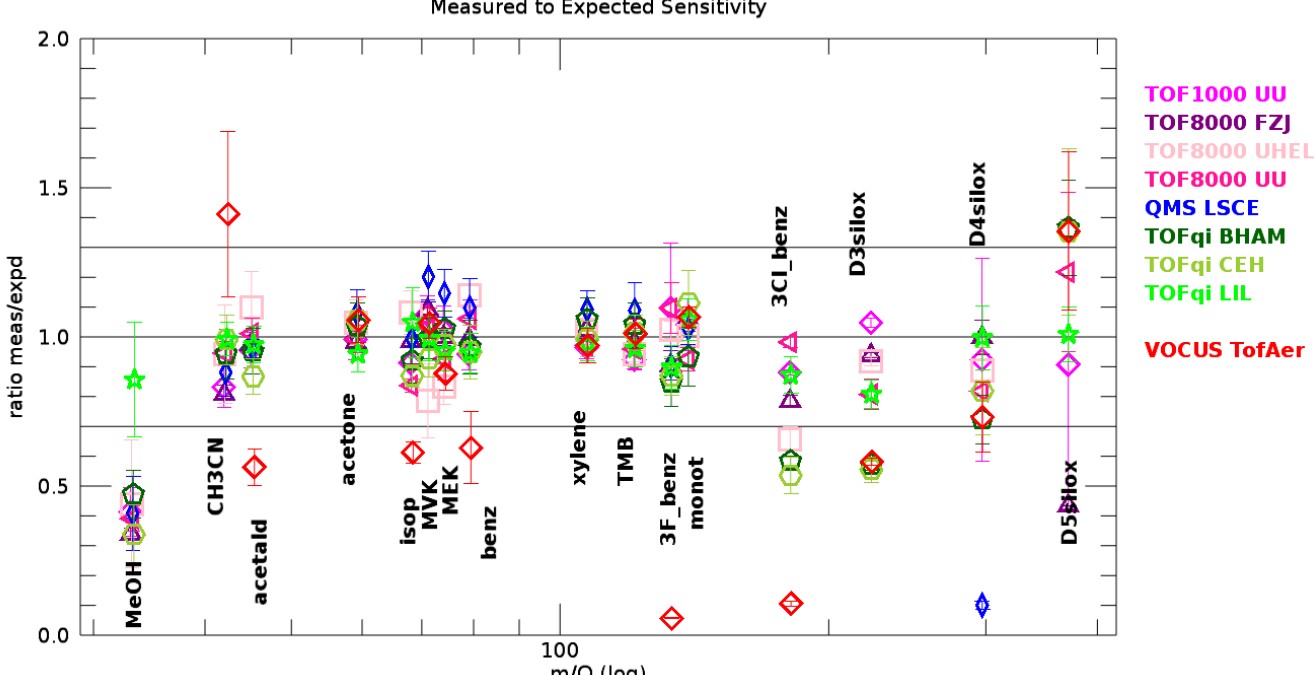

**Figure 5.** The ratio between measured and expected sensitivities as retrieved from dry injections that were not used for transmission retrievals. The data for most compounds and most instruments are well within +/- 30%. The error bars represent the standard deviation of all gas standard injections with dry $N_2$ or air as carrier gas, except those that were used to calculate the transmission . Compounds as in Figure 4.





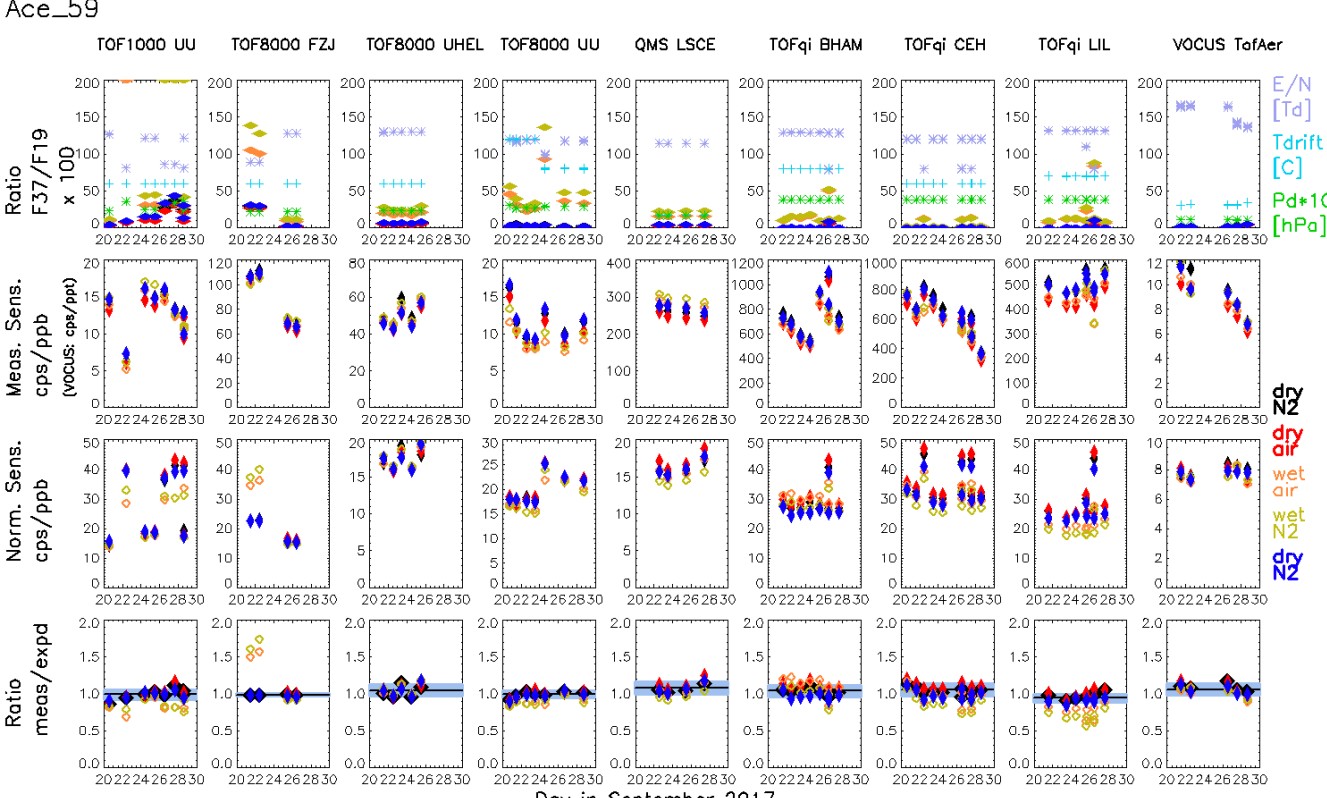

*Figure 6.* Summary for all measurements of acetone following our calibration protocol. Individual instruments are shown in the columns. The ***first-row*** panels show the ratio of primary ions $H_2OH_3O^+$ to $H_3O^+$ as well as operating conditions of the instruments (temperature, ℃, pressure in the drift tube, hPa, and *E/N*, Td, i.e. $10^{17}$ Vm$^2$). The ***second-row*** panels show the measured sensitivity of acetone for all instruments. The ***third-row*** panels show the normalized sensitivity, i.e. the measured sensitivity normalized to a transmission corrected primary ion signal (sum of $H_3O^+ + H_2OH_3O^+$) of $10^6$ counts per second. The ***fourth-row*** charts show the ratio of the measured to expected sensitivity. The median ratio and the standard deviation of all ratios using dry carrier gas are plotted as black vertical line and grey shade, respectively. The colours and markers represent the different carrier gases. Humidified injections are depicted with open markers (orange and yellow-green for air and nitrogen, respectively); filled markers depict calibrations in dry carrier gas (black, red, and blue for nitrogen, air, and nitrogen, respectively).





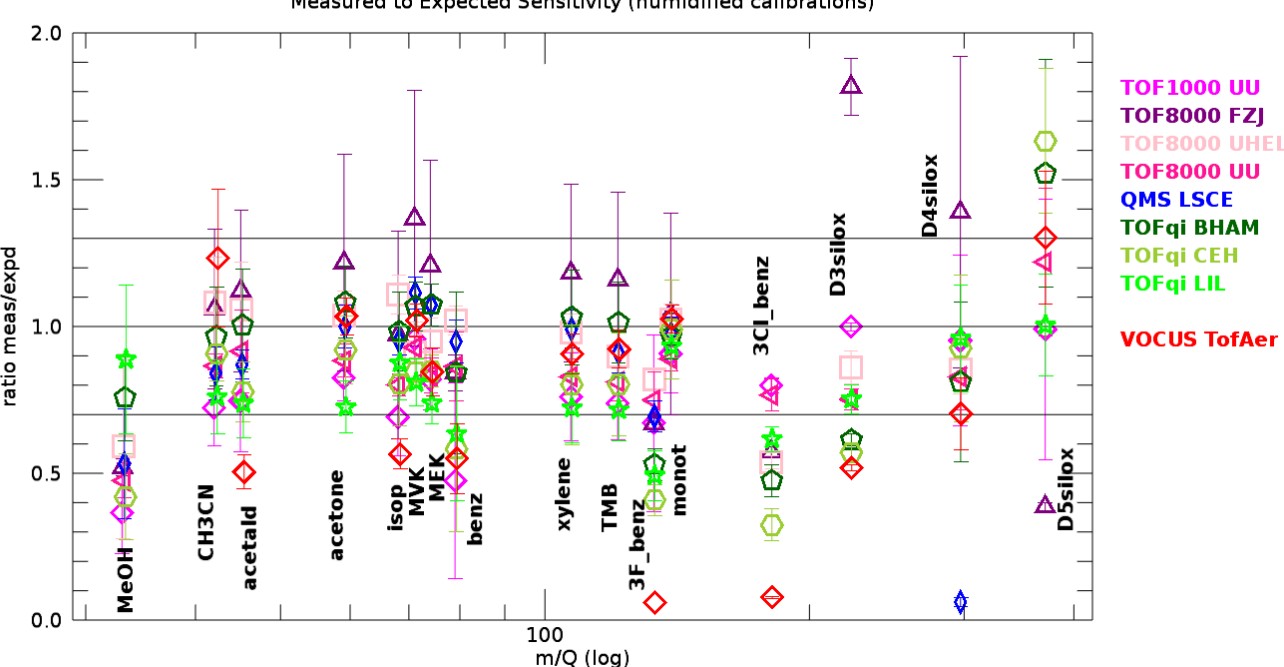

*Figure 7.* Measured versus expected sensitivities retrieved from humidified injections for all compounds in the gas standard and all PTR-MS instruments. The error bars represent the standard deviation of all gas standard injections with humidified $N_2$ or air as carrier gas. Compounds as in Figure 4.

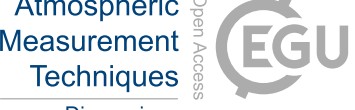



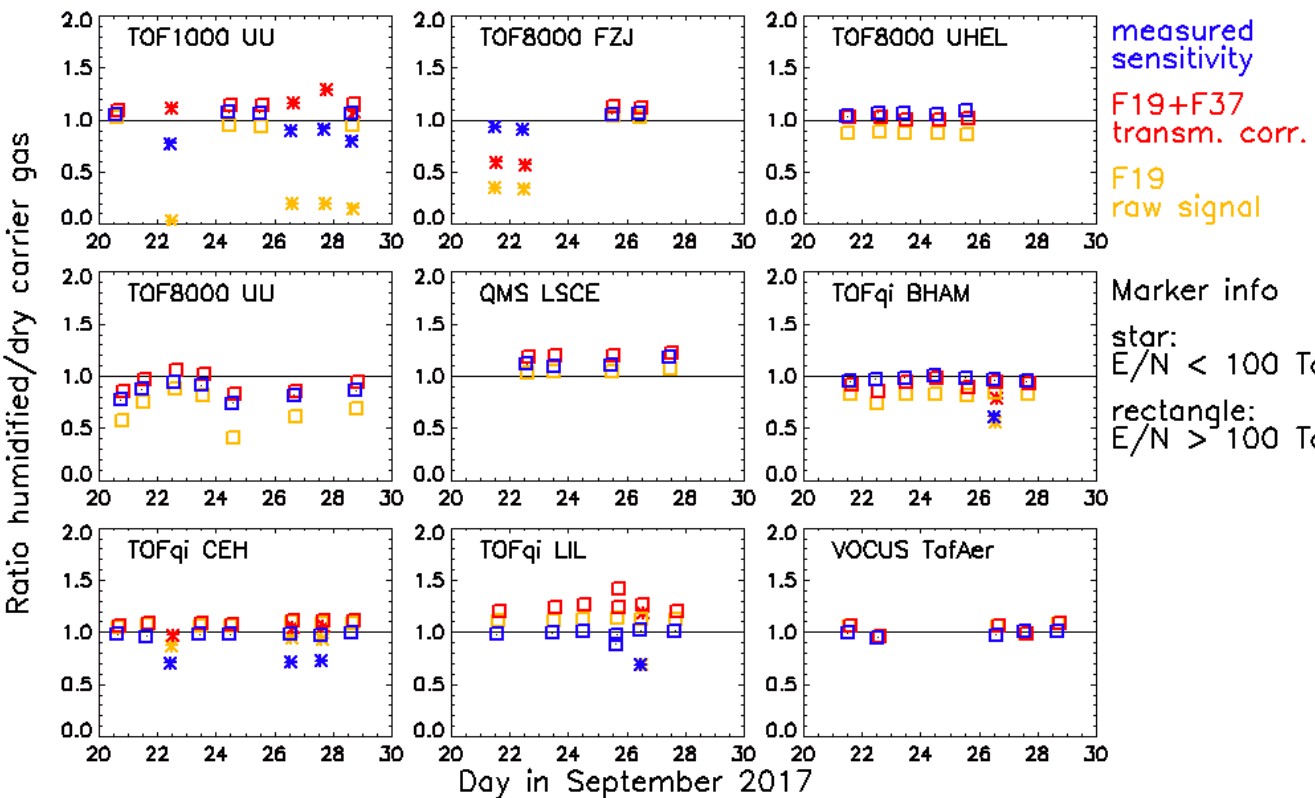

**Figure 8.** Comparison of gas standard injections using humidified and dry carrier gas. The ratios obtained for humidified versus dry injections are shown for (i) the mean measured sensitivity of a core set of compounds (blue), (ii) the transmission corrected primary ion signal (red), and (iii) the raw $H_3O^+$ signal (yellow). The symbols separate measurements done at *E/N* levels above (rectangles) or below (stars) 100 Td.



## Appendix A. Calculation of expected sensitivities for the PTR3

In the PTR3, the reaction time of primary ions is defined by the gas flow through its reaction chamber, estimated to be 3.5 ± 0.5 ms. Together with a pressure of 65 mbar, primary ions undergo approximately 1000 times more collisions with the analyte gas compared to all other instruments described in this paper. While this concept overall leads to greatly enhanced

sensitivities, it also complicates quantification: back reactions (de-protonation) of the form $RH^+ + H_2O \rightarrow H_3O^+ + R$ are observed for a broader range of species. The PTR3 is operated at a reduced electric field strength of 60 ± 5 Td, therefore the primary ion distribution is dominated by water clusters. Operation at higher E/N would push the equilibrium further towards de-protonation, which is not desirable. The observed mass spectrum does not necessarily reflect the true cluster distribution in the reaction chamber, since it is influenced by electric fields in the transfer region towards the mass spectrometer (de

Gouw et al, 2003). We use equilibrium constants experimentally obtained by Lau et al (1982) to obtain the primary ion cluster distribution. The effective ion temperature is calculated following de Gouw et al. (2003), using drift velocities $v_{drift}$ calculated with ion mobilities from Dotan et al. (1976) for individual hydronium water clusters:

$$T_{eff} = \frac{2}{3k_B}\left(\frac{1}{2} \cdot \frac{(m_{ion} + m_B) \cdot m_{H2O}}{m_{ion} + m_{H2O}} \cdot v_{drift}^2 + \frac{3}{2}k_B T\right)$$

$m_{ion}$, $m_B$ and $m_{H2O}$ are the masses of individual water clusters, the mean molecular mass of the buffer gas (air) and the

molecular mass of water, respectively; $k_B$ is the Boltzmann constant.

The forward reaction rate constants $k_f$ are calculated using the parametrization of T. Su (1994) and within ±30 % of the reaction rate constants in Table S3. To account for potential equilibrium conditions due to aforementioned back reactions, we apply the following formula to calculate sensitivities:

$$S_{expd} = \sum_{n=0}^{6} 10^{-9} \cdot k_f \cdot \frac{p_{react}}{k_B T_{react}} \cdot I(H_3O^+ \cdot (H_2O)_n) \cdot \frac{1}{k_r \cdot c_{H2O}} \cdot (1 - e^{-k_r c_{H2O} t_{react}})$$

Where $10^{-9} \frac{p_{react}}{k_B T_{react}}$ corresponds to a volume mixing ratio of 1 ppbv; $I(H_3O^+ \cdot (H_2O)_n)$ is the ion current of the n-th hydronium water cluster in counts per second and $c_{H2O}$ is the water vapor number density. The reverse reaction rate constant $k_r$ is calculated via

$$k_r = k_f \cdot e^{-\frac{PA(R) - PA(H2O)}{E_{cm}}}$$

With $PA(R)$ and $PA(H_2O)$ being the proton affinities of molecule R and water, respectively. $E_{cm}$ is the center-of-mass

kinetic energy between the protonated molecule $RH^+$ and water vapor, calculated according to de Gouw et al. (2003).

Using retrieved transmissions(example shown in Figure A2), this method leads to good agreement between expected and measured sensitivities for acetone, methyl ethyl ketone (MEK), methyl vinyl ketone (MVK), octamethylcycletetrasiloxane (D4) and decamethylcyclopentasiloxane (D5), as shown in Figure A1. However, sensitivities for methanol, acetonitrile, isoprene and α–pinene are overestimated. Smith et al. (2001) showed that Isoprene only reacts with $H_3O^+$ and $H_3O^+ \cdot$

$(H_2O)_n$ (n = 0 and n = 1) - by limiting the available primary ions for ionization to these two species, expected sensitivities agree with the measured values within uncertainties. Similar adjustments had to be applied for methanol, acetonitrile and α-





pinene (n < 3, respectively). The error bars of the expected sensitivities in Figure A1 show that careful calibrations for these compounds are necessary, since the values are sensitive to operational conditions (humidity, reduced electric field and temperature).



**Figure A1.** Measured (blue points) and expected sensitivities for the PTR3 plotted versus the respective proton affinities (PA). Orange crosses represent expected sensitivities taking into account the back reaction with water vapor and assuming ionization of the respective species with all water clusters $H_3O^+ \cdot (H_2O)_n$ ; $n = 0\text{-}6$; The black crosses are adjusted to react only with the lowest water clusters $n = 0 - k$ with $k = 2$ for methanol, acetonitrile and α-pinene; $k = 1$ for Isoprene. The error bars represent the combined uncertainties of the expected values resulting from uncertainties of the water vapor partial pressure ($1.2 \pm 0.6$ mbar), temperature ($30 \pm 5$ °C), reduced electric field strength ($60 \pm 5$ Td) in the reaction chamber. The proton affinities of D4 and D5 siloxanes are unknown, shown for reference only and assumed to be > 200 kcal/mol.



*Figure A2:* Retrieved transmissions for the Harvard PTR3, using a reduced subset of compounds: Acetone, MVK, MEK,
D3-, D4- and D5 siloxanes.

