# Peer review of "Validity and limitations of simple reaction kinetics to calculate concentrations of organic compounds from ion counts in PTR-MS"

_Atmospheric Measurement Techniques, 2018_

## Referee Comment (RC1) · Anonymous Referee #2 · 6 Mar 2019

This paper reports measurements by unique calibration protocol on 10 different PTR-MS instruments and comparison with a reaction kinetic model. The authors conclude that the kinetic model accurately predicts the sensitivities if no artefacts interfere and the instruments were operated at E/N levels above 100 Td. However, I have a question about the kinetic model, in terms of how to think the reaction time for quadrupole-type drift tube (comment 1). Also, kinetics for the PTR3 instrument should be discussed more deeply (comments 2 and 3). My major concerns should be addressed before accepted for publication.

Major concerns: 1. Page 7, line 20: The authors have used equation 11 to calculate

the residence time for all the instruments except for "PTR3 HAR". I have a question here: Can this equation be applied to the case of the instruments with quadrupole-type drift tubes (e.g., TOFqi and Vocus)? Eq. 11 is originally for static lens ion optics in which ions move linearly, and therefore, the drift tube length can be used directly for the calculation of the time. In contrast, ions move spirally in the quadrupole-type drift tube. This means ion trajectories are quite long compared to the drift tube length. The authors should clarify how they consider this matter.

2. Page 9, line 12 etc.: The authors state that primary ions undergo approximately 1000 times more collisions with the analyte gas in the PTR3 instrument. Taking into account this matter and Figure A1, I think that the following proton transfer reaction between analyte gas (R) and ion (RH+) should be considered, as well as de-protonation (RH+ + H2O → H3O+ + R). R1H+ + R2 → R2H+ + R1 (if PA(R2) > PA(R1), PA: proton affinity) The authors should discuss how the analyte sensitivities are affected by this reaction in case of the PTR3 instrument. Additionally, the density of water molecules in the drift tube should be discussed if de-protonation is considered.

3. Figure A1 in page 27: The measured sensitivities for isoprene and alpha-pinene are quite low compared to the expected ones. Do these results mean that excess fragmentation occurs in the PTR3 instrument? In the case of the PTR3 instrument, is there a possibility that CID (collision-induced dissociation)-like fragmentation occus? It would be good if the authors could suggest any reasons for quite low sensitivities for isoprene and alpha-pinene in the PTR3.

Minor concerns: 4. Page 27, line 19: "t_react" should be defined here or in the main text.

---

## Referee Comment (RC2) · Anonymous Referee #3 · 23 Apr 2019

**Review of Validity and limitations of simple reaction kinetics to calculate concentrations of organic compounds from ion counts in PTR-MS**

**AMT-2018-446**

**Holzinger et al.**

**Summary:**

A large number of PTR-MS instruments were calibrated using the same calibration system and gas standard cylinders. The reproducibility of the derived sensitivities is reported and compared between different instruments. Further discussion explores the effect of sample matrix and instrument operating condition on the calibration reproducibility. A method for estimating the mass-dependence of ion transmission through a PTR instrument is presented. In general it seems that PTR measurements are highly reproducible. Various instruments have different sensitivities and response to humidity, depending on instrument model and ionization technique, as expected. Mass-dependence transmission can significantly change the detected ion count rate and should be taken into consideration when calculating sensitivities of uncalibrated compounds.

This kind of intercomparison is a valuable exercise and I think these data should be available to the PTR community. The discussion of transmission curve is interesting and novel. However, on a whole the presented data and discussion are rather limited, and the paper is somewhat unfocused. The research objective stated in the title of the paper cannot be addressed without a larger number of compounds, an exploration of functional-group dependence, and assessment of the effect of fragmentation. Some important details of methods and data treatment are omitted. I can see this work as a valuable component of a larger study, for example an intercomparison of ambient data from the PICAB campaign or a more robust presentation of the transmission curve analysis. The paper would require major revisions before being publishable.

**Major comments:**

The purpose of the paper is a little unclear. Is it to evaluate the agreement between different PTR instruments or evaluate long-term stability? The title suggests it is to evaluate a method of determining sensitivity for uncalibrated compounds. If so, effect of fragmentation and wall-interactions should be considered.

The linear relationship between proton-transfer rate constant and sensitivity for select group of simple compounds has been reported since some time. This aspect is not particularly novel and does not require so much discussion.

The consideration of mass-dependent transmission is interesting and important for a robust calculation of sensitivity but is weakened by the very small number of compounds used to derive the curve. The degree of uncertainty in the derived transmission curve is not quantified. It would be of interest to describe how much the transmission curves vary between instruments, whether the curves can be wholly explained by known effects with ion optics, tof duty cycle, etc; and how much we can expect the transmission curve to vary in a single instrument given different instrument settings and typical campaign variability.

**Specific comments:**

The introduction is missing references to several highly relevant papers including methods to relate reaction kinetics to concentrations (Sekimoto et al. IJMS 2018; Cappellin et al. EST 2012, etc), a reference to the source of the kinetic rate constants In Table S3, and recent reviews that include assessments of PTR-MS precision and accuracy (e.g. Yuan et al. Chem. Rev. 2017). Additionally the introduction needs some more background about mass-dependent transmission, why it is important to consider, and previous methods used to determine the transmission curve in CIMS instrumentation. Relatedly there should be some information about the different hardware components of the system and how they're expected to affect transmission in different masses ranges; this is necessary to introduce Section 2.2.1.

Section 2: Can you please provide an overview of the instruments and the relevant differences among them?

Page 3 section 2.1 Here are missing some details about how the calibration system operates. Are the calibration compounds injected into the sample loop as pure gases, or is a pre-diluted sample of a standard cylinder collected in the sample loop and later released into a secondary dilution stream? What is the advantage of using this system compared to diluting a constant stream of calgas with known flow into a dilution stream (dynamic dilution)? What is the material of the calibration system? Were wall-loss effects considered?

Page 3 line 25: Was a comparison of the two successive calibration cylinders conducted, to ensure consistency?

Page 3 line 28: Is there any information on long-term stability of siloxanes in cylinders? Monoterpenes are known to be not especially stable long-term. When were the cylinders produced?

Fig. 2: The many subfigures and small text make this figure difficult to read and grasp the main point. Why is dry nitrogen included twice?

Eq. 1: what is f(M)- the detected signal? In cps or ncps? For the time-of-flight instruments, is it corrected for the duty-cycle of the ToF extraction region? When adding together the sensitivities of fragmenting compounds, which mass is used for M: the parent mass, the fragment, or an average?

Were the forms of equations 1,2, 3 determined empirically or are these based on some knowledge of relevant ion physics?

Page 6 line 4. It would be helpful to have a figure in the main text showing an example optimized fit of Eq1, Eq2, Eq3 to the six compounds, as well as the final retrieved transmission from Eq4 compared to real data (e.g. Figure S2 as a publication-quality graphic).

Page 7: Is the effect of the much higher temperature in the drift tube (compared to ambient) on the kinetic rate constants considered?

Page 7 line 17: Not clear if normalizing factor "N" includes correction for transmission; says so in page 6 line 22 but not here?

Page 8 line 19: what is meant by "steeper" transmission?

Page 11 line 12: isn't this in contradiction to earlier statement about PTR3; the high sensitivity is due to long reaction time?

Page 13 bottom half: It is well understood that humidity affects PTR sensitivity and there are well established methods for correcting the sensitivities of conventional PTR instruments. Were corrections for humidity not applied?

Page 12 Line 29: probably because the primary ion distribution is shifted towards mz 37, which is heavier than mz 19 and therefore transmitted with higher efficiency

Figures 5-7: Why is PTR3 HAR not included in these figures?

Figure 6: Is the ratio of m37:19 corrected for transmission effects? If not, this is not a particularly useful point of comparison, because it reflects the downstream ion optics rather than the actual conditions present in the drift tube.

Figure 6: The sensitivity of some instruments, even after normalization, seems to be quite unstable from day-to-day. Can you comment on this?

**Technical corrections:**

Footnotes would be better placed in the main text as part of the methods explanation.

Eq. 10; note that this is only valid if reagent ions are negligibly depleted

---

## Author Comment (AC1) · 7 Jul 2019

**Final Author's response to reviewer comments on "Validity and limitations of simple reaction kinetics to calculate concentrations of organic compounds from ion counts in PTR-MS" by Rupert Holzinger et al.**

We thank referee #2 for their insightful and high-quality comments on our manuscript. In the following we address their comments point by point. The referee comments are copied to this document in *blue* font.

**Report #1(referee #2):**

*Major concerns:*
*1. Page 7, line 20: The authors have used equation 11 to calculate the residence time for all the instruments except for "PTR3 HAR". I have a question here: Can this equation be applied to the case of the instruments with quadrupole-type drift tubes (e.g., TOFqi and Vocus)? Eq. 11 is originally for static lens ion optics in which ions move linearly, and therefore, the drift tube length can be used directly for the calculation of the time. In contrast, ions move spirally in the quadrupole-type drift tube. This means ion trajectories are quite long compared to the drift tube length. The authors should clarify how they consider this matter.*

**Answer:** Note that the TOFqi instruments apply an static electric field along the reaction chamber, so eq. 11 applies. The quadrupole in these instruments is used in the ion-transfer region between the reaction chamber and the mass analyzer – a region where no ionization should occur. This situation is indeed different for the Vocus instrument, where a RF field is applied across the entire length of the reaction chamber. However, the residence time is determined by the static electrical field in the direction of the ion drift. The RF-components are perpendicular to this direction and suppress the diffusive broadening of the ion beam. The RF components have insignificant influence on the residence time and in eq. 11 we used only the static electric field in the drift direction.

We added following footnote to clarify this in the manuscript: *"Note that we used only the static electric field in the drift direction in Eq. 11 to calculate the residence time for the Vocus. The RF-components are perpendicular to the drift direction and have little influence on the residence time in the reactor."*

*2. Page9,line12etc.: The authors state that primary ions undergo approximately 1000 times more collisions with the analyte gas in the PTR3 instrument. Taking into account this matter and Figure A1, I think that the following proton transfer reaction between analyte gas (R) and ion (RH$^+$) should be considered, as well as de-protonation (RH$^+$ + H$_2$O→H$_3$O$^+$ + R). R1H$^+$ + R2→R2H$^+$ + R1 (if PA(R2) > PA(R1), PA: proton affinity) The authors should discuss how the analyte sensitivities are affected by this reaction in case of the PTR3 instrument. Additionally, the density of water molecules in the drift tube should be discussed if de-protonation is considered.*

De-protonation through collision with a high PA compounds can become relevant if their concentration is in the proximity of 100 nmol/mol or higher. Practically this is not an issue that needs to be considered, because such high concentrations are outside the operation range of the PTR3 instrument (the primary ions would be titrated almost completely). The concentration of water vapor certainly needs to be accounted for in an advanced reaction kinetic model for PTR3. However, this is beyond the scope of this work, which is evaluating the inter-comparability of standard PTR-MS systems.

*3. Figure A1 in page 27: The measured sensitivities for isoprene and alpha-pinene are quite low compared to the expected ones. Do these results mean that excess fragmentation occurs in the PTR3 instrument? In the case of the PTR3 instrument, is there a possibility that CID (collision-induced dissociation)-like fragmentation occurs? It would be good if the authors could suggest any reasons for quite low sensitivities for isoprene and alpha-pinene in the PTR3.*

E/N in the PTR3 is low, so fragmentation is not expected. Figure A1 also shows "adjusted expected" sensitivities for these compounds, which are in better agreement with the measured

sensitivities. As we explained in appendix A (just above Fig A1), these adjustments are based on Smith et al. (2001), who found that isoprene is not ionized by higher water clusters.

*Minor concerns: 4. Page 27, line 19: "t_react" should be defined here or in the main text.*

Done. We expanded the sentence starting at line 20 as follows:

"*Where* $10^{-9} \frac{p_{react}}{k_B T_{react}}$ *corresponds to a volume mixing ratio of 1 ppbv (with $p_{react}$ and $T_{react}$ corresponding to the pressure and temperature in the reactor, respectively, and $k_B$ to the Boltzmann constant); $I(H_3O^+ \cdot (H_2O)_n)$ is the ion current of the n-th hydronium water cluster in counts per second; $k_r$, $t_{react}$, and $c_{H2O}$ correspond to the forward reaction rate constant, the residence time in the reactor, and the water vapor number density, respectively.*"

---

## Author Comment (AC2) · 7 Jul 2019

**Final Author's response to reviewer comments on "Validity and limitations of simple reaction kinetics to calculate concentrations of organic compounds from ion counts in PTR-MS" by Rupert Holzinger et al.**

We thank referee #3 for their insightful and high-quality comments on our manuscript. In the following we address their comments point by point. The referee comments are copied to this document in *blue* font.

**Report #2(referee #3):**

*Major comments:*

*The purpose of the paper is a little unclear. Is it to evaluate the agreement between different PTR instruments or evaluate long-term stability? The title suggests it is to evaluate a method of determining sensitivity for uncalibrated compounds. If so, effect of fragmentation and wall-interactions should be considered.*

As the title clearly states the purpose of this work, which is testing the validity of simple reaction kinetics to calculate VOC concentrations for a larger suite of PTR-MS instruments, all operated under field conditions. A method to constrain the transmission of the mass analyzer is a fundamental requirement to analyze the validity of the kinetic model and therefore this is a central part of the presented work.

Wall interactions and fragmentation are issues that certainly deserve careful attention, but these issues were not the focus of the present work. We want to refer to the EUROCHAMP PTR-MS Intercomparison Campaign that took place in May 2019 at the HELIOS chamber in France that was designed to address questions associated with wall interactions, humidity, fragmentation, and the detection of uncalibrated oxidation products.

*The linear relationship between proton-transfer rate constant and sensitivity for select group of simple compounds has been reported since some time. This aspect is not particularly novel and does not require so much discussion.*

It is true that the relationship has been reported, but at the same time the limitations of kinetic approach have been highlighted, and the reliability of uncalibrated PTR-MS measurements has been questioned. This is the first study that tested the kinetic approach on many different PTR-MS instruments operated under common field conditions. We think that this is an important work that will encourage the PTR-MS community to explore the scientific wealth of untargeted (thus uncalibrated) measurements.

*The consideration of mass-dependent transmission is interesting and important for a robust calculation of sensitivity but is weakened by the very small number of compounds used to derive the curve. The degree of uncertainty in the derived transmission curve is not quantified. It would be of interest to describe how much the transmission curves vary between instruments, whether the curves can be wholly explained by known effects with ion optics, tof duty cycle, etc; and how much we can expect the transmission curve to vary in a single instrument given different instrument settings and typical campaign variability.*

We do not agree with the notion that the transmission is retrieved by a very small number of compounds. Actually, there are no big gaps between 33 and 137 Th, above that there were less compounds but the coverage until 371 Th is unique! We agree that more compounds are desirable especially above 200 Th. In our manuscript we discuss that due to this deficiency the transmission is less well constrained above 200 Th, which can be seen for example in the bottom, middle chart of Figure 3.

The expected behavior of the transmission curve is described in section 2.2.1. For example, the general characteristic of a TOF-MS (the square root relationship) is preserved in the medium mass

range. At lower masses the ion intensities are reduced by ion optics between reactor and mass analyzer. The optics act as a high-masses-pass filter because focusing techniques are more efficient for heavier ions. In some instruments the effect is intentionally enhanced to reduce the primary ion intensities and detector ageing. Detector ageing or insufficient TOF-MS tuning often acts as low-masses-pass filter in the high mass range. In answering the reviewer's question we conclude that we have fair knowledge to explain the behavior of the transmission curve.

Long term stability of the transmission curve depends on stable operating condition and on minimizing/correcting detector ageing. These conditions were met for 2 instruments (TOF8000 UHEL, and QMSLSCE, see section 3.1 and Figure 3).

*The introduction is missing references to several highly relevant papers including methods to relate reaction kinetics to concentrations (Sekimoto et al. IJMS 2018; Cappellin et al. EST 2012, etc), a reference to the source of the kinetic rate constants In Table S3, and recent reviews that include assessments of PTR-MS precision and accuracy (e.g. Yuan et al. Chem. Rev. 2017). Additionally the introduction needs some more background about mass-dependent transmission, why it is important to consider, and previous methods used to determine the transmission curve in CIMS instrumentation. Relatedly there should be some information about the different hardware components of the system and how they're expected to affect transmission in different masses ranges; this is necessary to introduce Section 2.2.1.*

We extended the discussion and added the requested references. Following text has been added to the introduction at line 15 on page 2:

*"For example, Cappellin et al. (2012) demonstrated the quantitative properties of a PTR-TOF8000 instrument by assuming a theoretical transmission based on the duty-cycle in the time-of-flight mass analyser. However, new generation instruments substantially gained sensitivity by using advanced ion optics between reactor and mass analyser at the cost that the transmission of the whole system is less well constrained – especially in the lower mass range. In addition, poor tuning and/or ageing of the ion detection system can cause deviations from the expected behaviour at the high mass range. So, fast and robust methods for retrieving the transmission are needed for quantitative measurements. Another requirement is the knowledge of the reaction rate constant for proton transfer between protonated water and the compound to be quantified. Proton transfer reactions typically occur at collisional rates, which can be calculated using quantum chemical methods (e.g. Su, 1994; Zhao and Zhang, 2004). Sekimoto et al. (2017) developed a method to estimate the reaction rate constant from the molecular composition rather than from molecular polarizability and dipole moment. Such attempts are promising and may further increase the stand-alone quantitative capacity of PTR-MS by exploiting parameters that are directly measured (i.e. the molecular composition of the ion to be quantified)."*

*Section 2: Can you please provide an overview of the instruments and the relevant differences among them?*

This information is provided in Table 1, which is already referenced in the introduction (page 3, line 5).

*Page 3 section 2.1 Here are missing some details about how the calibration system operates. Are the calibration compounds injected into the sample loop as pure gases, or is a pre-diluted sample of a standard cylinder collected in the sample loop and later released into a secondary dilution stream? What is the advantage of using this system compared to diluting a constant stream of calgas with known flow into a dilution stream (dynamic dilution)? What is the material of the calibration system? Were wall-loss effects considered?*

On page 3, lines 24-29, provide detailed information about the gas standard cylinders we used, so 'a pre-diluted sample of a standard cylinder' is used. We will provide this information earlier in the section (see below). We added the missing information concerning materials, wall loss, and the advantages of using the sample loop approach (see below).

In the second sentence of section 2.1 we added the information in **bold** letters:

*"A small flow (~10 mL/min) of carrier gas transports the content of the sample loop (**i.e. a multi-component gas standard containing approximately 1 μmol/mol per compound in N₂**) to a T-connection where it is mixed into a larger flow (0.2-2 L/min) of dry or humidified carrier gas."*

We added following sentences to section 2.1:

*"In order to avoid wall loss, the sample loop, 6-port valve, and dilution system are operated at approximately 80˚C, and all materials in contact with the gas standard are either stainless steel with sulfinert® coating (Restek Inc.), or Teflon PFA. The sample loop approach allows to perform full calibrations very quickly (within 1-2 seconds). Multiple calibrations in a row are feasible to assess the quality of the calibrations directly and/or to explore different operating conditions (e.g. humidified or dry gas, E/N settings etc.)."*

*Page 3 line 25: Was a comparison of the two successive calibration cylinders conducted, to ensure consistency?*

This was mainly for practical reasons: in the beginning of the campaign the NPL standard was not yet available. On the other hand, the A&R standard contained 2 more compounds in the high mass range.

*Page 3 line 28: Is there any information on long-term stability of siloxanes in cylinders? Monoterpenes are known to be not especially stable long-term. When were the cylinders produced?*

The A&R was produced in 2016 and the NPL standard just before the campaign. The stability of the compounds in the NPL standard has been re-confirmed by NPL after the campaign. Comparing the 2 standards we see no indication that the siloxanes were unstable in the A&R cylinder.

*Fig. 2: The many subfigures and small text make this figure difficult to read and grasp the main point. Why is dry nitrogen included twice?*

As explained in section 2.1 (and in the Figure caption) the protocol started AND ended with a set of 10 dry N₂ injections.

*Eq. 1: what is f(M)- the detected signal? In cps or ncps? For the time-of-flight instruments, is it corrected for the duty-cycle of the ToF extraction region? When adding together the sensitivities of fragmenting compounds, which mass is used for M: the parent mass, the fragment, or an average?*

$f_1(M)$ is the first of the 3 functions that approximate the transmission and it is a function of m/Q. The 'TOF duty-cycle' is the reason why we expect a square root relationship, i.e. the parameter *a* in Eq. 1 is expected to be around 0.5. We used the parent mass for adding the sensitivities. We are aware that this procedure underestimates the sensitivity of isoprene if there was no fragmentation, however, typically less than 20% if isoprene fragmented and therefore this limitation did not affect the retrieval algorithm significantly. To clarify this we changed the last sentence of this paragraph to:

*"For fragmenting compounds (e. g. isoprene) we added the measured sensitivity of the fragment to the sensitivity of the protonated ion (used in the algorithm)."*

*Were the forms of equations 1,2, 3 determined empirically or are these based on some knowledge of relevant ion physics?*

As explained above the forms were chosen to describe the general behavior of a TOF mass analyzer (Eq.1), the characteristics of the ion optics in the transfer region between reactor and TOF (Eq.2), and potential issues with tuning/ageing of the ion detection system (Eq. 3).

*Page 6 line 4. It would be helpful to have a figure in the main text showing an example optimized fit of Eq1, Eq2, Eq3 to the six compounds, as well as the final retrieved transmission from Eq4 compared to real data (e.g. Figure S2 as a publication-quality graphic).*

First we want to point out that the all compounds in the standard are used to retrieve the transmission (more than six!). We were considering this a lot, but eventually decided to show the overview of all transmissions and all instruments in the main publication (Figure 3) because this is of general interest. Figures S2 and S3 provide more detailed technical information on the performance of the algorithm and we feel that the supplement is an appropriate location.

*Page 7: Is the effect of the much higher temperature in the drift tube (compared to ambient) on the kinetic rate constants considered?*

Yes.

*Page 7 line 17: Not clear if normalizing factor "N" includes correction for transmission; says so in page 6 line 22 but not here?*

Yes. As stated $S_N$ uses the factor "N", $S_m$ does not.

*Page 8 line 19: what is meant by "steeper" transmission?*

Steeper means a larger parameter 'a' (Eq.1). We clarified steeper and flatter in this paragraph by adding (larger parameter 'a' in Eq.1) and (smaller parameter 'a' in Eq.1), respectively.

*Page 11 line 12: isn't this in contradiction to earlier statement about PTR3; the high sensitivity is due to long reaction time?*

Yes, PTR3 is different and the statement does not refer to PTR3. The discussion on page 11, line 12, is about features in Figure 6, which does not show PTR3 data.

*Page 13 bottom half: It is well understood that humidity affects PTR sensitivity and there are well established methods for correcting the sensitivities of conventional PTR instruments. Were corrections for humidity not applied?*

Humidity effects are well known and corrections have been suggested. However, these effects are complex and depend on compound properties, energetics in the reactor, and the partial water pressure in the reactor. The latter two define the distribution of water clusters and may produce complex pathways for protonation. All three parameters control possible de-protonation pathways which are strongly dependent on humidity. All available corrections are empirical and depend on instrument and operating conditions. Therefore we did not apply any corrections, but we defined the operating conditions in which the humidity effects are relatively minor.

*Page 12 Line 29: probably because the primary ion distribution is shifted towards mz 37, which is heavier than mz 19 and therefore transmitted with higher efficiency*

The transmission is corrected for (red data in Figure 8). Moreover, the explanation suggested by the reviewer is ruled out by the fact that also the $H_3O^+$ signal increases under humid conditions (yellow data in Fig 8).

*Figures 5-7: Why is PTR3 HAR not included in these figures?*

The reason is that the simple kinetic model does not describe the conditions in the PTR3. Therefore we excluded PTR3 from this discussion, but dedicated an appendix to the specialties of this instrument.

*Figure 6: Is the ratio of m37:19 corrected for transmission effects? If not, this is not a particularly useful point of comparison, because it reflects the downstream ion optics rather than the actual conditions present in the drift tube.*

Yes. We now specified this in the Figure caption.

*Figure 6: The sensitivity of some instruments, even after normalization, seems to be quite unstable from day-to-day. Can you comment on this?*

This reflects different operating conditions such as E/N, pressure, and temperature in the reactor.

Footnotes would be better placed in the main text as part of the methods explanation.

We prefer to keep the footnotes. They provide extra information while the flow of the main text remains focused.

*Eq. 10; note that this is only valid if reagent ions are negligibly depleted*

Thanks for pointing this out. We now specified this requirement.

---

## Referee Report (RR1)

**Review of** Validity and limitations of simple reaction kinetics to calculate concentrations of organic compounds from ion counts in PTR-MS
**AMTD,** amt-2018-446
Holzinger et al.

**Summary:**

The authors calibrated a large number of different PTR-MS instruments and used this information to derive mass-dependent transmission curves, and to evaluate a few other controls on sensitivity, including the effect of humidity. The purpose is to determine if sensitivity of a particular PTR-MS instrument to a particular VOC can be estimated, when the VOC rate constant and transmission through the instrument are known. The authors present a method for determining the transmission as a function of m/Q. Overall the expected and measured sensitivities agree quite well, if transmission is taken into account. A major remaining uncertainty is the effect of high concentrations of water cluster ($H_3O^+(H_2O)_n$).

**Major comments:**

The manuscript is improved from the previous version. The work is not particularly novel, as the effect of humidity on sensitivity and the reagent ion distribution, the mass-dependence of transmission, and recommendations for PTR operating conditions have been addressed previously in many other publications. The paper mainly focusses on transmission effects, so I am not sure that the analysis is comprehensive enough truly evaluate sensitivity calculation as promised in the title. However the number of different PTR instruments compared is unprecedented and this paper could be a useful resource for other PTR operators.

I have a few major questions:

1) Equation (1) seems to be a correction for the duty-cycle of ion extraction into the ToF region of a mass analyzer. For a ToF mass spectrometer *a* should be exactly equal to 0.5. Why not apply the known duty-cycle correction directly, before fitting equations (2) and (3)?

2) The purpose of the paper is to evaluate the agreement between sensitivity calculated from reaction kinetics, and measured sensitivities. Any number of things could cause measured sensitivities to deviate from expected sensitivities, including mass-dependent transmission, differences in rate constant with $H_3O^+$ vs $H_2O*H_3O^+$ or larger water cluster back-reaction, fragmentation, uncertainty in rate constant, and effect of contaminant reagent ions (e.g. $O_2^+$).

The paper focuses almost exclusively on determination of mass-dependent transmission and states that quantitative detection is possible within 30%. Is the 30% uncertainty entirely attributable to difficulty in determining the transmission function? Is the largest source of error the transmission curve, reaction with water clusters, or some other effect?

**Specific/minor comments:**

Page 9 line 11: "We find that typically 'flatter' transmissions (smaller parameter 'a' in Eq.1) were retrieved when the instruments were (deliberately) operated at lower E/N (thin lines in Figure 3)."

This seems a little strange because Eq.1 is a function of the mass analyzer, not the drift tube – why would the drift tube conditions affect the extraction duty-cycle? (or anything downstream)? Is it that the transmission as a function of mass is not actually changed, but is rather an artifact of lower reaction rate for certain compounds used to determine the transmission curve? If so, then this suggests that

benzene etc. should be excluded from determination of the transmission curve, or that only dry air and/or high E/N should be used.

Page 13 Line 22. Likely this is because m37 is consistently transmitted with higher efficiency than m19. Therefore if most reagent ions begin their journey into the downstream optics as m37, and then at some later point decluster to produce m19, the observed m19 will be higher during humidified measurements.

Page 14 line 13: A citation is needed for this sentence: "in traditional PTR-MS applications that focus on volatile organic compounds fragmentation of compounds is the exception rather than the rule"

Figure 3: It would be much easier to compare the transmissions of various instruments if the y-axis scale were the same in each subplot.

---

## Author Response (AR2)

**Final Author's response to reviewer comments on "Validity and limitations of simple reaction kinetics to calculate concentrations of organic compounds from ion counts in PTR-MS" by Rupert Holzinger et al.**

We thank referee #3 for their insightful and high-quality comments on our manuscript. In the following we address their comments point by point. The referee comments are copied to this document in *blue* font.

**Report #1(Anonymous Referee #3, Aug 13, 2019):**

*Major questions:*
*1) Equation (1) seems to be a correction for the duty-cycle of ion extraction into the ToF region of a mass analyzer. For a ToF mass spectrometer a should be exactly equal to 0.5. Why not apply the known duty-cycle correction directly, before fitting equations (2) and (3)?*

The reviewer is correct that equation (1) accounts mostly for the duty cycle of time-of-flight mass analyzers. However, there are other parameters that can influence the transmission in, e.g. diffusion in the reactor, or ion guide systems in the transfer region. The goal was to provide a general algorithm that can be used for a wide range of different designs – including quadrupole mass analyzers, where *a=0.5* is not expected.

*2) The purpose of the paper is to evaluate the agreement between sensitivity calculated from reaction kinetics, and measured sensitivities. Any number of things could cause measured sensitivities to deviate from expected sensitivities, including mass-dependent transmission, differences in rate constant with H3O+ vs H2O\*H3O+ or larger water cluster back-reaction, fragmentation, uncertainty in rate constant, and effect of contaminant reagent ions (e.g. O2+).*

*The paper focuses almost exclusively on determination of mass-dependent transmission and states that quantitative detection is possible within 30%. Is the 30% uncertainty entirely attributable to difficulty in determining the transmission function? Is the largest source of error the transmission curve, reaction with water clusters, or some other effect?*

The 30% uncertainty is not entirely attributable to the transmission function. We actually think that the major fraction is du to the other processes that are mentioned by the reviewer, namely differences in rate constant with $H_3O^+$ vs $H_2O^*H_3O^+$ or larger water cluster back-reaction, fragmentation, and uncertainty in the rate constant. The individual contributions cannot be disentangled with the present set of measurements. However, the measurements using humidified carrier gas clearly shows the limitations of the simple kinetic model. Whether the accuracy of the transmission, water cluster effects, or uncertainty in the reaction rate constant are the largest source of error depends on the operating conditions and on the compounds that are considered. The effects of water clusters (i.e. lower rate constant and de-protonation) are extensively discussed in Section 3.3. The important result of this work is that we clearly define operating conditions that limit the total error to less than 30% for most compounds.

*Specific/minor comments:*

*Page 9 line 11: "We find that typically 'flatter' transmissions (smaller parameter 'a' in Eq.1) were retrieved when the instruments were (deliberately) operated at lower E/N (thin lines in Figure 3)."*

*This seems a little strange because Eq.1 is a function of the mass analyzer, not the drift tube – why would the drift tube conditions affect the extraction duty-cycle? (or anything downstream)? Is it that the transmission as a function of mass is not actually changed, but is rather an artifact of lower reaction rate for certain compounds used to determine the transmission curve? If so, then this suggests that benzene etc. should be excluded from determination of the transmission curve, or that only dry air and/or high E/N should be used.*

The transmission is also influenced by the conditions in the drift tube, e.g. by (mass dependent) diffusion, and the ion optics between the drift tube and the mass analyzer. The referee is right that the transmission of the mass analyzer does not change by changing E/N, however, the transmission of the whole system, which is relevant here, may change. Note that the performance of the algorithm is not compromised if one compound does not comply with the simple model, such as benzene with high abundance of water clusters.

*Page 13 Line 22. Likely this is because m37 is consistently transmitted with higher efficiency than m19. Therefore if most reagent ions begin their journey into the downstream optics as m37, and then at some later point decluster to produce m19, the observed m19 will be higher during humidified measurements.*

This is an interesting idea but we do not know if it is likely at all. Following the referee we would need to assume that most reagent ions pass the "high-masses-pass" filter as m37. After the filter there would have to be ion-molecule collisions that cause the de-clustering so that in the TOF region m19 is 'only' reduced by the duty-cycle. This could indeed explain the observed behavior. However, we have no exact information on the pressure and physical dimensions in this region, so we feel that it is too speculative to be added to the manuscript.

*Page 14 line 13: A citation is needed for this sentence: "in traditional PTR-MS applications that focus on volatile organic compounds fragmentation of compounds is the exception rather than the rule"*

We added the reference Lindinger et al. 1998.

*Figure 3: It would be much easier to compare the transmissions of various instruments if the y-axis scale were the same in each subplot.*

One important purpose of this Figure is to show the variability of transmission retrievals for the individual instruments. For many instruments this would be much harder to see if we used the same y-scale. Therefore we prefer to keep the current scales.